# The MR-Base platform supports systematic causal inference across the human phenome

Gibran Hemani[1†]*, Jie Zheng[1†]*, Benjamin Elsworth[1†], Kaitlin H Wade[1], Valeriia Haberland[1], Denis Baird[1], Charles Laurin[1], Stephen Burgess[2], Jack Bowden[1], Ryan Langdon[1], Vanessa Y Tan[1], James Yarmolinsky[1], Hashem A Shihab[1], Nicholas J Timpson[1], David M Evans[1,3], Caroline Relton[1], Richard M Martin[1], George Davey Smith[1], Tom R Gaunt[1‡]*, Philip C Haycock[1‡]*

[1]Medical Research Council (MRC) Integrative Epidemiology Unit, Population Health Sciences, Bristol Medical School, University of Bristol, Bristol, United Kingdom; [2]Department of Public Health and Primary Care, University of Cambridge, Cambridge, United Kingdom; [3]University of Queensland Diamantina Institute, Translational Research Institute, Brisbane, Australia

*For correspondence:
g.hemani@bristol.ac.uk (GH);
jie.zheng@bristol.ac.uk (JZ);
tom.gaunt@bristol.ac.uk (TRG);
philip.haycock@bristol.ac.uk (PCH)

†These authors contributed equally to this work
‡These authors also contributed equally to this work

Competing interests: The authors declare that no competing interests exist.

**Abstract** Results from genome-wide association studies (GWAS) can be used to infer causal relationships between phenotypes, using a strategy known as 2-sample Mendelian randomization (2SMR) and bypassing the need for individual-level data. However, 2SMR methods are evolving rapidly and GWAS results are often insufficiently curated, undermining efficient implementation of the approach. We therefore developed MR-Base (http://www.mrbase.org): a platform that integrates a curated database of complete GWAS results (no restrictions according to statistical significance) with an application programming interface, web app and R packages that automate 2SMR. The software includes several sensitivity analyses for assessing the impact of horizontal pleiotropy and other violations of assumptions. The database currently comprises 11 billion single nucleotide polymorphism-trait associations from 1673 GWAS and is updated on a regular basis. Integrating data with software ensures more rigorous application of hypothesis-driven analyses and allows millions of potential causal relationships to be efficiently evaluated in phenome-wide association studies.
DOI: https://doi.org/10.7554/eLife.34408.001

## Introduction

Inferring causal relationships between phenotypes is a major challenge and has important implications for understanding the aetiology of disease processes. The potential for phenome-wide causal inference has increased markedly over the past 10 years due to two major advances. The first is the continuing success of large scale genome-wide association studies (GWAS) in identifying robust genetic associations (*Visscher et al., 2017*). The second is the development of statistical methods for causal inference that exploit the principles of Mendelian randomization (MR) using GWAS summary data (*Davey Smith and Ebrahim, 2003*; *Davey Smith and Hemani, 2014*; *Zhu et al., 2016*; *Pierce and Burgess, 2013*). Genetic data for MR can, however, be difficult to access, while MR methods are evolving rapidly and can be difficult to implement for non-specialists. To address the need for more systematic curation and application of complete GWAS summary data and MR methods, we have developed MR-Base (http://www.mrbase.org): a platform that integrates a database of thousands of GWAS summary datasets with a web interface and R packages for automated causal inference through MR. Following an extended introduction on the uses and sources of GWAS

**eLife digest** Our health is affected by many exposures and risk factors, including aspects of our lifestyles, our environments, and our biology. It can, however, be hard to work out the causes of health outcomes because ill-health can influence risk factors and risk factors tend to influence each other. To work out whether particular interventions influence health outcomes, scientists will ideally conduct a so-called randomized controlled trial, where some randomly-chosen participants are given an intervention that modifies the risk factor and others are not. But this type of experiment can be expensive or impractical to conduct.

Alternatively, scientists can also use genetics to mimic a randomized controlled trial. This technique – known as Mendelian randomization – is possible for two reasons. First, because it is essentially random whether a person has one version of a gene or another. Second, because our genes influence different risk factors. For example, people with one version of a gene might be more likely to drink alcohol than people with another version. Researchers can compare people with different versions of the gene to infer what effect alcohol drinking has on their health.

Every day, new studies investigate the role of genetic variants in human health, which scientists can draw on for research using Mendelian randomization. But until now, complete results from these studies have not been organized in one place. At the same time, statistical methods for Mendelian randomization are continually being developed and improved. To take advantage of these advances, Hemani, Zheng, Elsworth et al. produced a computer programme and online platform called "MR-Base", combining up-to-date genetic data with the latest statistical methods.

MR-Base automates the process of Mendelian randomization, making research much faster: analyses that previously could have taken months can now be done in minutes. It also makes studies more reliable, reducing the risk of human error and ensuring scientists use the latest methods. MR-Base contains over 11 billion associations between people's genes and health-related outcomes. This will allow researchers to investigate many potential causes of poor health. As new statistical methods and new findings from genetic studies are added to MR-Base, its value to researchers will grow.

DOI: https://doi.org/10.7554/eLife.34408.002

summary data, and the principles and assumptions behind MR, we describe how to implement MR analyses using MR-Base, how to interpret results and provide a thorough overview of potential limitations. In an applied example, we demonstrate the functionality of MR-Base through an MR study of low density lipoprotein (LDL) cholesterol and coronary heart disease (CHD). We also demonstrate how the integration achieved by MR-Base supports a wide range of applications, including phenome-wide association studies (PheWAS) to identify potential sources of horizontal pleiotropy, and for performing hypothesis-free MR to gain insight into impacts of interventions. These applications demonstrate how integrating data and analytical tools enable novel insights that would previously have been technically and practically challenging to achieve.

## GWAS summary data

GWAS summary data, the non-disclosive results from testing the association of hundreds of thousands to millions of genetic variants with a phenotype, have been routinely collected and curated for several years (*Welter et al., 2014*; *Li et al., 2016*; *Beck et al., 2014*) and are a valuable resource for dissecting the causal architecture of complex traits (*Pasaniuc and Price, 2017*). Accessible GWAS summary data are, however, often restricted to 'top hits', that is, statistically significant results, or tend to be hosted informally in different locations under a wide variety of formats. For other studies, summary data may only be available 'on request' from authors. Complete summary data are currently publicly accessible for thousands of phenotypes but to ensure reliability and efficiency for systematic downstream applications they must be harvested, checked for errors, harmonised and curated into standardised formats. GWAS summary data are useful for a wide variety of applications, including MR, PheWAS (*Millard et al., 2015*; *Denny et al., 2010*), summary-based transcriptome-wide (*Gusev et al., 2016*) and methylome-wide (*Richardson et al., 2017*; *Hannon et al., 2017a*)

association studies and linkage disequilibrium (LD) score regression (*Bulik-Sullivan et al., 2015*; *Zheng et al., 2017b*).

## Mendelian randomization

MR (*Davey Smith and Ebrahim, 2003*; *Davey Smith and Hemani, 2014*) uses genetic variation to mimic the design of randomised controlled trials (RCT) (although for interpretive caveats see *Holmes et al., 2017*). Let us suppose we have a single nucleotide polymorphism (SNP) that is known to influence some phenotype (the exposure). Due to Mendel's laws of inheritance and the fixed nature of germline genotypes, the alleles an individual receives at this SNP are expected to be random with respect to potential confounders and causally upstream of the exposure. In this 'natural experiment', the SNP is considered to be an instrumental variable (IV), and observing an individual's genotype at this SNP is akin to randomly assigning an individual to a treatment or control group in a RCT (*Figure 1a*). To infer the causal influence of the exposure, one calculates the ratio between the SNP effect on the outcome over the SNP effect on the exposure. If there are many independent IVs available for a particular exposure, as is often the case, causal inference can be strengthened (*Johnson, 2012*). Here, we consider each SNP to mimic an independent RCT and we can adapt tools developed for meta-analysis (*Bowden et al., 2017a*) to combine the results obtained from each of the SNPs, giving an overall causal estimate that is better powered (*Bowden et al., 2017a*).

Crucially, MR can be performed using results from GWAS, in a strategy known as 2-sample MR (2SMR) (*Pierce and Burgess, 2013*). Here, the SNP-exposure effects and the SNP-outcome effects are obtained from separate studies. With these summary data alone, it is possible to estimate the causal influence of the exposure on the outcome. This has the tremendous advantage that causal inference can be made between two traits even if they aren't measured in the same set of samples, enabling us to harness the statistical power of pre-existing large GWAS analyses. Due to the flexibility afforded by the 2SMR strategy, MR can be applied to 1000s of potential exposure-outcome associations, where 'exposure' can be very broadly defined, from gene expression and proteins to more complex traits, such as body mass index and smoking.

While MR avoids certain problems of conventional observational studies (*Davey Smith and Ebrahim, 2001*), it introduces its own set of new problems. MR is predicated on exploiting 'vertical' pleiotropy, where a SNP influences two traits because one trait causes the other (*Davey Smith and Hemani, 2014*). It is crucial to be aware of the assumptions and limitations that arise due to this model (*Haycock et al., 2016*). The main assumptions (*Figure 1b*) are: the instrument associates with the exposure (IV assumption 1); the instrument does not influence the outcome through some pathway other than the exposure (IV assumption 2); and the instrument does not associate with confounders (IV assumption 3). The IV1 assumption is easily satisfied in MR by restricting the instruments to genetic variants that are discovered using genome-wide levels of statistical significance and replicated in independent studies. The other two assumptions are impossible to prove, and, when violated, can lead to bias in MR analyses. Violations of the IV2 assumption can be introduced by 'horizontal' pleiotropy where the SNP influences the outcome through some pathway other than the exposure. Such effects can manifest in various different patterns (*Figure 1c–f*). When multiple independent instruments are available it is possible to perform sensitivity analyses that attempt to distinguish between horizontal and vertical pleiotropy and return causal estimates adjusted for the former (*Bowden et al., 2016a*; *Bowden et al., 2015*; *Hartwig et al., 2017b*). To improve reliability of causal inference, MR results should be presented alongside sensitivity analyses that make allowance for various potential patterns of horizontal pleiotropy. Further details on the design and interpretation of Mendelian randomization studies can be found in several existing reviews (*Davey Smith and Hemani, 2014*; *Haycock et al., 2016*; *Swerdlow et al., 2016*; *Holmes et al., 2017*; *Zheng et al., 2017a*). A glossary of terms can be found in *Supplementary file 1F*.

## Model

In this section we describe how to use MR-Base to conduct MR analyses (*Figure 2*). The data required to perform the analysis can be described as a 'summary set' (*Hemani et al., 2017a*), where the genetic effects for a set of instruments are available for both the exposure and the outcome. To create a summary set we select appropriate instruments, obtain the effect estimates for those

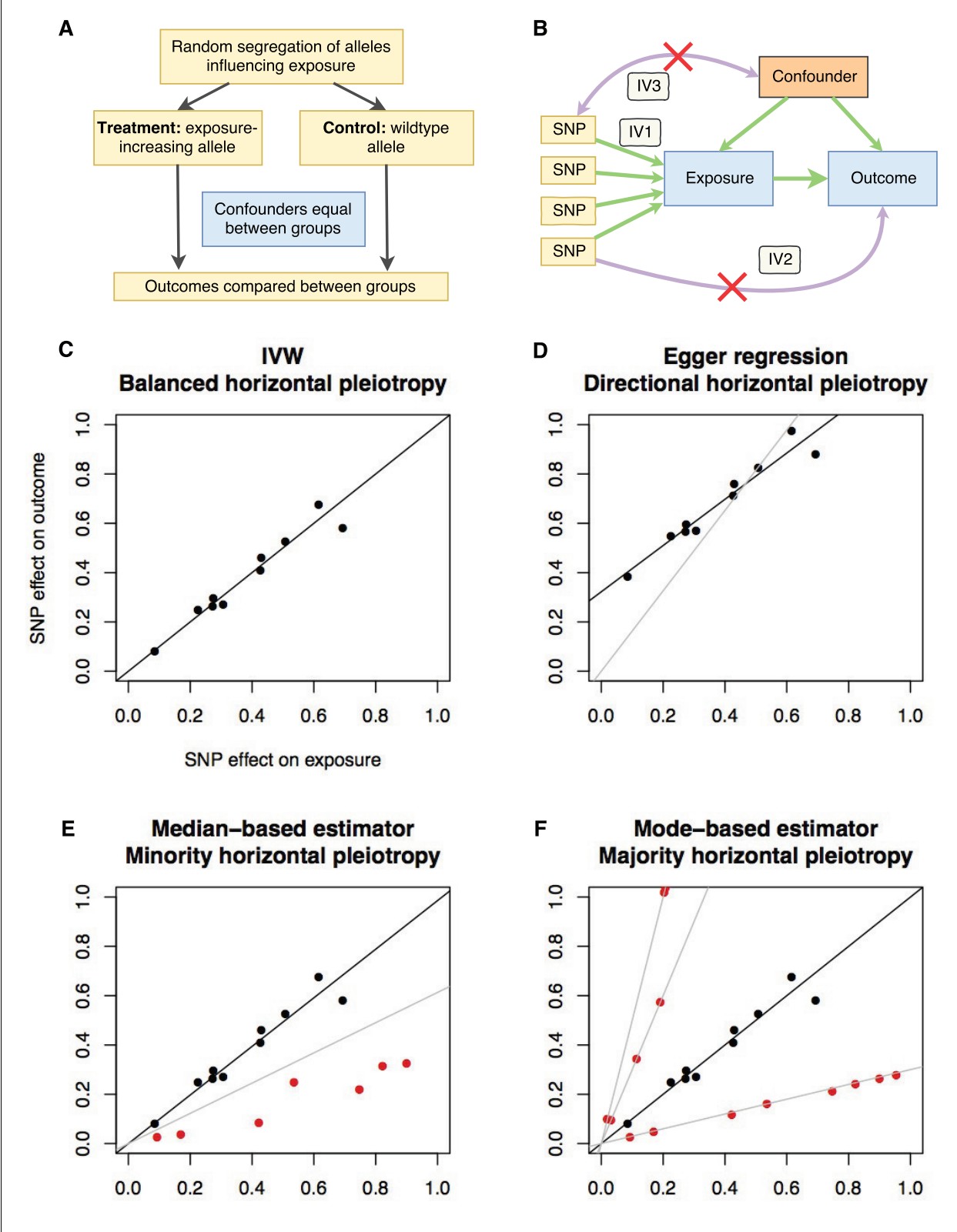

**Figure 1.** Principles and assumptions behind Mendelian randomization. (A) Diagram illustrating the analogy between Mendelian randomization (MR) and a randomised controlled trial. (B) A directed acyclic graph representing the MR framework. Instrumental variable (IV) assumption 1: the instruments must be associated with the exposure; IV assumption 2: the instruments must influence the outcome only through the exposure; IV assumption 3: the instruments must not associate with measured or unmeasured confounders. (C-F) Scatter plots demonstrating the relationship between the instrumental

*Figure 1 continued on next page*

*Figure 1 continued*

single nucleotide polymorphism (SNP) effects on the exposure against their corresponding effects on the outcome. The slope of the regression is the estimate of the causal effect of the exposure on the outcome. (C) If there is no violation of the IV2 assumption (no horizontal pleiotropy), or the horizontal pleiotropy is balanced, an unbiased causal estimate can be obtained by inverse-variance weighted (IVW) linear regression, where the contribution of each instrumental SNP to the overall effect is weighted by the inverse of the variance of the SNP-outcome effect. Fixed and random effects IVW approaches are available (the slopes from both approaches are identical but the variance of the slope is inflated in the random effects model in the presence of heterogeneity between SNPs). (D) If there is a tendency for the horizontal pleiotropic effect to be in a particular direction, then constraining the slope to go through zero will incur bias (grey line). Egger regression relaxes this constraint by allowing the intercept to pass through a value other than zero, returning an unbiased effect estimate if the instrument-exposure and pleiotropic effects are uncorrelated, also known as the InSIDE (Instrument Strength Independent of Direct Effect) assumption (*Bowden et al., 2015*). Pleiotropic effect here refers to the effect of the instrument on the outcome that is not mediated by the exposure. (E) If the majority of the instruments are valid (black points), with some invalid instruments (red points), the median based approach will provide an unbiased estimate in the presence of unbalanced horizontal pleiotropy (black line), whereas IVW linear regression will provide a biased estimate (grey line). In addition, the median-based estimator does not require the InSIDE assumption of the Egger approach. (F) If a group of SNPs influences the outcome through a particular pathway other than the exposure (i.e. the SNPs are horizontally pleiotropic) then that group of SNPs will return consistently biased estimates. Clustering SNPs based on their estimates (grey lines) is possible with the mode-based estimator. The cluster with the largest weight (black line) is selected as the final causal estimate. The causal estimate from the mode-based estimator is unbiased if the SNPs contributing to the largest cluster are valid instruments.

DOI: https://doi.org/10.7554/eLife.34408.003

instruments for the exposure and the outcome, and harmonise the effects so that they reflect the same allele. We can then perform MR analyses using the summary set. These steps are supported by the database of GWAS results and R packages ('TwoSampleMR' and 'MRInstruments') curated by MR-Base and the following R packages curated by other researchers: 'MendelianRandomization' (*Yavorska and Burgess, 2017*), 'RadialMR' (*Bowden et al., 2017b*), 'MR-PRESSO' (*Verbanck et al., 2018*) and 'mr.raps' (*Zhao et al., 2018*). The statistical methods and R packages accessible through MR-Base are updated on a regular basis.

## Obtaining instruments

Instruments are characterised as SNPs that reliably associate with the exposure, meaning they should be obtained from well-conducted GWAS, typically involving their detection in a discovery sample at a GWAS threshold of statistical significance (e.g. $p < 5 \times 10^{-8}$) followed by replication in an independent sample. The minimum data requirements for each SNP are effect sizes ($\beta_x$), standard errors ($\sigma_x$) and effect alleles. Also useful are sample size, non-effect allele and effect allele frequency.

### Sources

There are several data sources that can be used in MR-Base (*Figure 3*) to define exposure and outcome traits (the number of traits is updated on a regular basis):

1.  The MR-Base database comprises complete GWAS summary data for hundreds of traits (*Figure 3* and *Supplementary file 1A*). By 'complete' we mean all SNPs reported in a GWAS analysis, with no exclusions on the basis of a p-value threshold for association with the target trait of interest. It is possible for the user to extract the top-hits from this data source using their own criteria (e.g. strength of p-value). Alternatively, potential instruments can be obtained from the MRInstruments package, which includes independent SNP-trait associations from the database with p-value < 5e-8.
2.  Quantitative trait loci (QTL) studies performed on DNA methylation (*Gaunt et al., 2016*), gene expression (*GTEx Consortium, 2015*), protein (*Deming et al., 2016*) and metabolite (*Shin et al., 2014*; *Kettunen et al., 2016*) levels generate hundreds to thousands of independent associations for thousands of traits. The MRInstruments R package contains hundreds of thousands of 'omic QTLs for ease of use within MR-Base.
3.  The NHGRI-EBI GWAS catalog (*Welter et al., 2014*) comprises 21,324 SNPs associated with 1628 complex traits and diseases. This list of potential instruments has been harmonised and formatted for ease of use within MR-Base within the MRInstruments R package.
4.  User provided data can also be used for analysis.

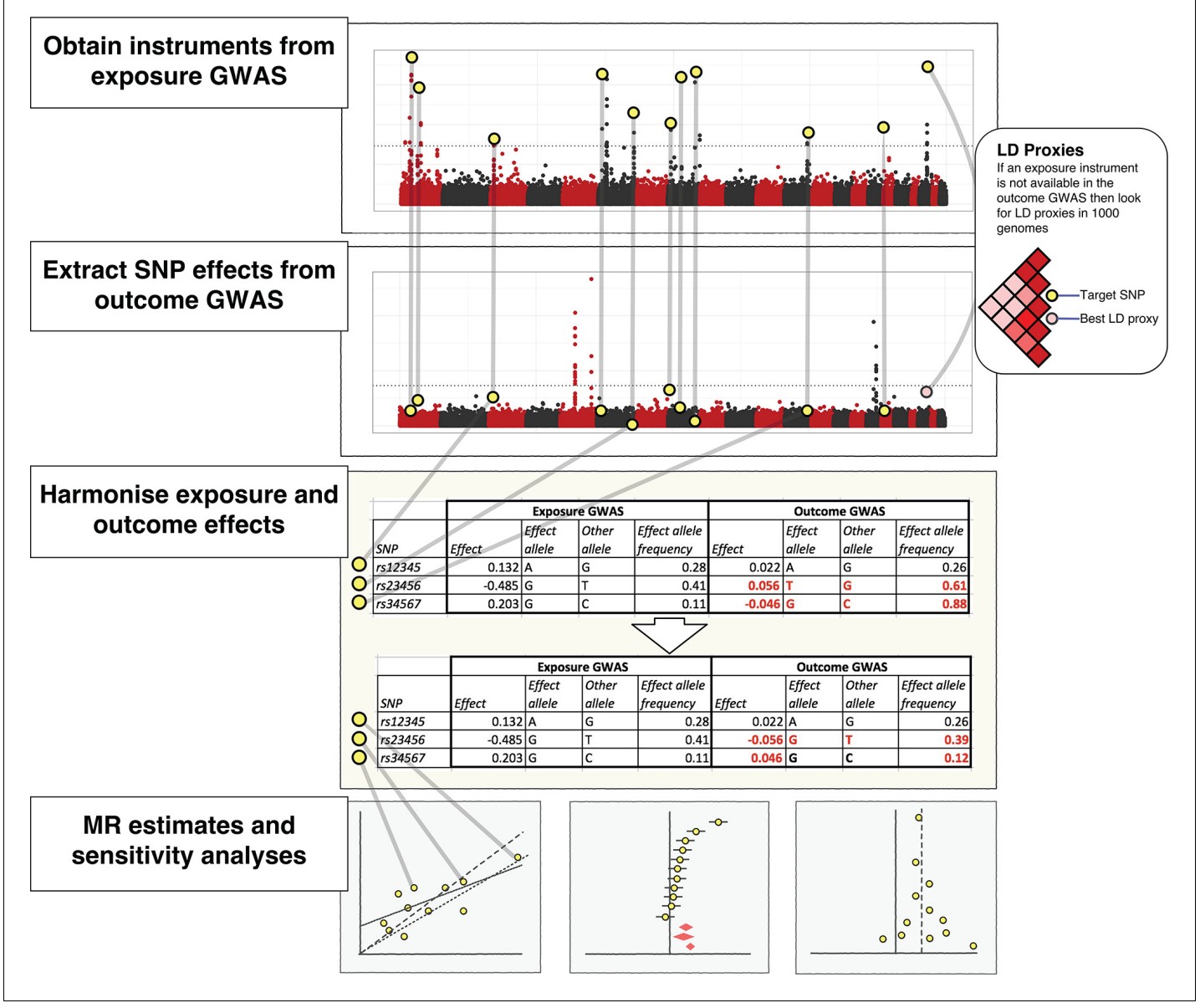

**Figure 2.** The practical steps for performing 2-sample Mendelian randomization (2SMR), as described in the Model section of the paper. The database of genome-wide association study results and R packages ('TwoSampleMR' and 'MRInstruments') curated by MR-Base support the data extraction, harmonisation and analysis steps required for 2SMR. Additional R packages for MR from other researchers are also accessible, including MendelianRandomization (*Yavorska and Burgess, 2017*), RadialMR ( *Bowden et al., 2017b*), MR-PRESSO (*Verbanck et al., 2018*) and mr.raps (*Zhao et al., 2018*). The available methods are updated on a regular basis.
DOI: https://doi.org/10.7554/eLife.34408.004

## Independence

It is important to ensure that instruments selected for an exposure are independent, unless measures are taken in the MR analysis to account for any correlation structures that arise through linkage disequilibrium. An efficient way to ensure that instruments are independent is to use clumping against a reference dataset of similar ancestry to the samples in which the GWAS was conducted. A clumping procedure has been implemented in MR-Base to automate the generation of independent instruments.

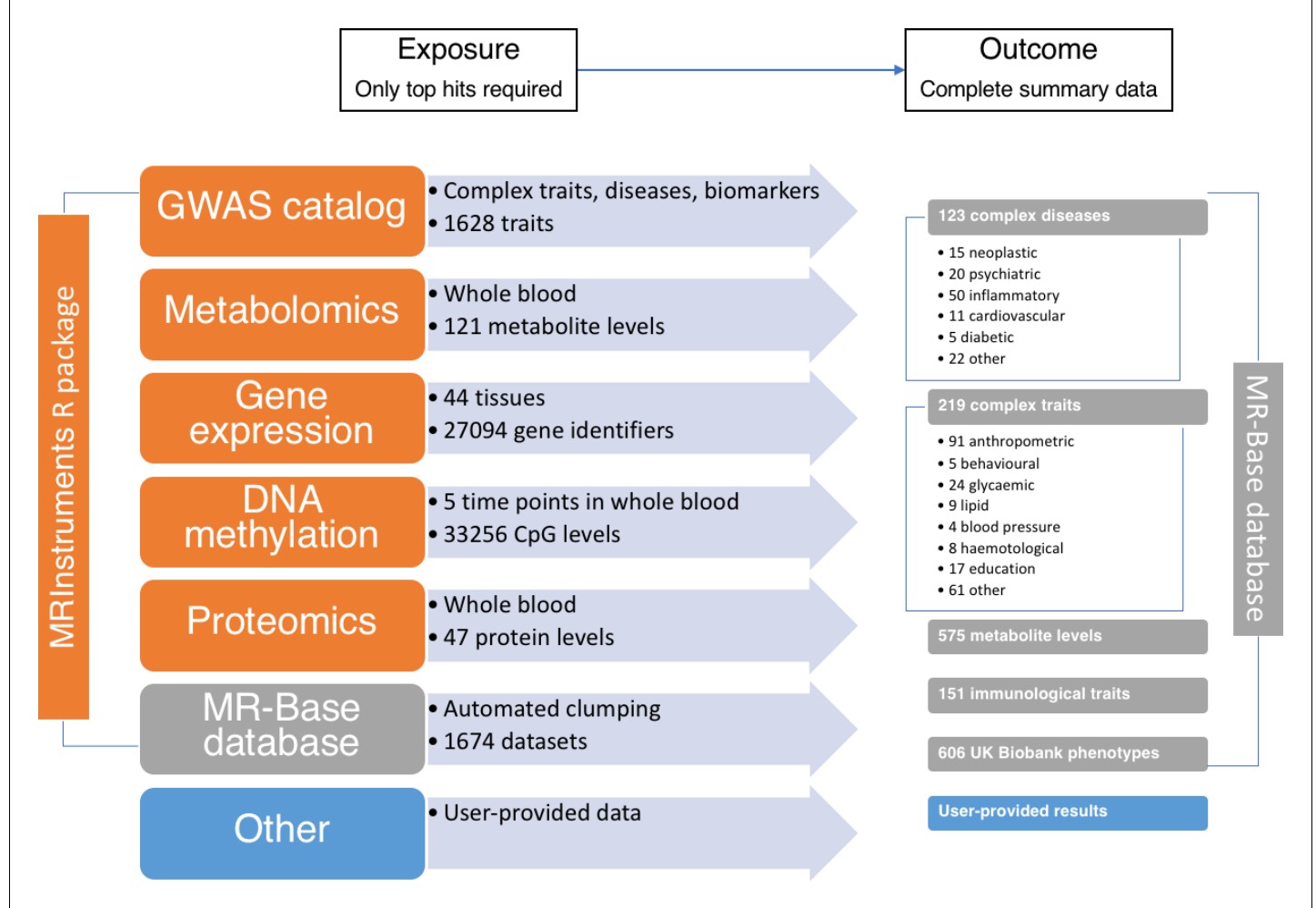

**Figure 3.** The data available through MR-Base and the possible exposure-outcome analyses that can be performed. Exposure traits can very broadly defined and may include molecular traits like gene expression, DNA-methylation, metabolites and proteins, as well as more complex traits, including cholesterol, body mass index, smoking and education. Further details on the traits with complete summary data can be found in *Supplementary file 1A*. The numbers reflect MR-Base in December 2017 and are updated on a regular basis.
DOI: https://doi.org/10.7554/eLife.34408.005

## Obtaining SNP effects on the outcome

In order to generate the summary set, the effects of each of the instruments on the outcome need to be obtained. This typically requires access to the entire set of GWAS results because it is unlikely that the instrumental SNPs for the exposure will be amongst the top hits of the outcome GWAS. As with the exposure data, the outcome data must contain at a minimum the SNP effects ($\beta_y$), their standard errors ($\sigma_y$) and effect alleles.

### LD proxies

If a particular SNP is not present in the outcome dataset then it is possible to use SNPs that are LD 'proxies' instead. Here, it is important to ensure that for any LD proxy used, the surrogate effect allele is the one in phase with the effect allele of the original target SNP. LD proxy lookups are automatically provided by MR-Base.

### Sources

There are two main sources that can be used (*Figure 3*):

1. The MR-Base database comprises complete GWAS summary data for hundreds of traits (*Supplementary file 1A*). Fast lookups for specific SNPs against specific traits can be performed. If a requested SNP is absent, then MR-Base automatically searches for LD proxies, estimated using data from the 1000 genomes project (*1000 Genomes Project Consortium et al., 2015*), and returns the corresponding data for the best proxy (*Figure 2*).
2. User provided complete GWAS summary data can be used with the R package.

## Harmonising exposure and outcome SNP effects

To generate a summary set, for each SNP we need its effect and standard error on the exposure and the outcome corresponding to the same effect alleles (*Hartwig et al., 2016*). This is impossible to generate if the effect alleles for the SNP effects in the exposure and outcome datasets are unknown. MR-Base uses knowledge of the effect alleles, and where necessary the effect allele frequencies, to automatically harmonise the exposure and outcome datasets. The following scenarios are considered:

### Wrong effect alleles
A SNP with (for example) effect/non-effect alleles G/T for the exposure and T/G for the outcome are harmonised by flipping the sign of the SNP-outcome effect.

### Strand issues
SNPs that are reported as (for example) G/T for the exposure summary dataset and C/A for the outcome dataset indicate a strand issue, where, for example, one study has reported the effect on the forward strand and the other on the reverse strand. In this case, the outcome alleles are flipped to match those of the exposure alleles, and effect alleles are then aligned.

### Palindromic SNPs
SNPs with A/T or G/C alleles are known as palindromic SNPs, because their alleles are represented by the same pair of letters on the forward and reverse strands, which can introduce ambiguity into the identity of the effect allele in the exposure and outcome GWASs. If reference strands are unknown, effect allele frequency can be used to resolve the ambiguity. For example, consider a SNP with alleles A and T, with a frequency of 0.11 for allele A in the exposure study and 0.91 in the outcome study. In addition, both studies have coded allele A as the effect allele and both are of European origin. The fact that allele A is the minor allele in the exposure study (frequency<0.5) and the major allele (>0.5) in the outcome study implies that the two studies have used different reference strands. To ensure that the effect sizes for the SNP reflect the same allele it is therefore necessary to switch the direction of the effect in either the exposure or outcome study (the default in MR-Base is to flip the direction of effect in the outcome study). Effect allele frequency may not, however, be a reliable indicator of reference strand when it is close to 0.5. This process has been described in more detail previously (*Hartwig et al., 2016*).

### Incompatible alleles
If a SNP has (for example) A/G alleles for the exposure and A/C alleles for the outcome, there is no combination of flipping that can reconcile these differences, and either there are build differences or there is an error in the data. In this instance the SNP is excluded from the analysis.

## Performing MR analysis

The generated summary set can now be analysed using a range of methods (summarised in *Supplementary file 1B* but new methods are added on a regular basis). The most basic way to combine these data is to use a Wald ratio where the estimated causal effect is

$$\beta_{MR} = \frac{\beta_y}{\beta_x}$$

and the standard error of the estimate is

$$\sigma_{MR} = \frac{\sigma_y}{\beta_x}$$

If there are multiple independent instruments for the exposure (as is typically the case for complex traits with well-powered GWAS), then our analysis can potentially improve in two major ways: first, the variance explained in the exposure, and therefore statistical power will improve; second, we can evaluate the sensitivity of the estimate to bias arising from violations of the IV2 assumption by assuming different patterns of horizontal pleiotropy. Sensitivity analyses are performed automatically by MR-Base.

## Inverse variance weighted MR

The simplest way to obtain an MR estimate using multiple SNPs is to perform an inverse variance weighted (IVW) meta analysis of each Wald ratio (*Johnson, 2012*), effectively treating each SNP as a valid natural experiment. Fixed effects IVW assumes that each SNP provides the same estimate or, in other words, none of the SNPs exhibit horizontal pleiotropy (or other violations of assumptions). Random effects IVW relaxes this assumption, allowing each SNP to have different mean effects, e.g. due to horizontal pleiotropy (*Bowden et al., 2017a*). This will return an unbiased estimate if the horizontal pleiotropy is balanced, i.e. the deviation from the mean estimate is independent from all other effects. Another way to conceptualise this result is as a weighted regression of the SNP-exposure effects against the SNP-outcome effects, with the regression constrained to pass through the origin, and with weights derived from the inverse of the variance of the outcome effects. MR-Base implements a random effects IVW model by default, unless there is underdispersion in the causal estimates between SNPs, in which case a fixed effects model is used. The estimates from the random and fixed effects IVW models are the same but the variance for the random effects model is inflated to take into account heterogeneity between SNPs.

## Maximum likelihood

An alternative strategy to the IVW approach is to estimate the causal effect by direct maximisation of the likelihood given the SNP-exposure and SNP-outcome effects and assuming a linear relationship between the exposure and outcome (*Pierce and Burgess, 2013*). Similar to the fixed effects IVW approach, the method assumes that the effect of the exposure on the outcome due to each SNP is the same, i.e. assumes there is no heterogeneity or horizontal pleiotropy. An unbiased estimate will be returned in the absence of horizontal pleiotropy or when horizontal pleiotropy is balanced (but the variance of the effect estimate will be overly precise in the latter case). An advantage of the method is that it may provide more reliable results in the presence of measurement error in the SNP-exposure effects.

## MR Egger analysis

Relaxing the IV2 assumption of 'no horizontal pleiotropy', MR-Egger (*Bowden et al., 2015*; *Bowden et al., 2016b*) adapts the IVW analysis by allowing a non-zero intercept, allowing the net-horizontal pleiotropic effect across all SNPs to be unbalanced, or directional. The method returns an unbiased causal effect even if the IV2 assumption is violated for all SNPs but assumes that the horizontal pleiotropic effects are not correlated with the SNP-exposure effects (this is known as the InSIDE assumption). Horizontal pleiotropy refers to the effects of the SNPs on the outcome not mediated by the exposure.

## Median-based estimator

An alternative approach is to take the median effect of all available SNPs (*Bowden et al., 2016a*; *Kang et al., 2014*). This has the advantage that only half the SNPs need to be valid instruments (i.e. exhibiting no horizontal pleiotropy, no association with confounders, robust association with the exposure) for the causal effect estimate to be unbiased. The weighted median estimate allows stronger SNPs to contribute more towards the estimate, and can be obtained by weighting the contribution of each SNP by the inverse variance of its association with the outcome.

## Mode-based methods

The mode-based estimator clusters the SNPs into groups based on similarity of causal effects, and returns the causal effect estimate based on the cluster that has the largest number of SNPs (*Hartwig et al., 2017b*). The mode-based method returns an unbiased causal effect if the SNPs within the largest cluster are valid instruments. Clustering is performed using a kernel density function that requires selecting a bandwidth parameter. The weighted mode introduces an extra element similar to IVW and the weighted median, weighting each SNP's contribution to the clustering by the inverse variance of its outcome effect.

## Diagnostics and sensitivity analyses

It is recommended that the methods described above are applied to all MR analyses and presented in publications to demonstrate sensitivity to different patterns of assumption violations. MR-Base also automatically performs the following further sensitivity analyses and diagnostics

### Heterogeneity tests

Heterogeneity in causal effects amongst instruments is an indicator of potential violations of IV assumptions (*Bowden et al., 2017a*). Heterogeneity can be calculated for the IVW and Egger estimates, and this can be used to navigate between models of horizontal pleiotropy (*Bowden et al., 2017a*).

### Leave-one-out analysis

To evaluate if the MR estimate is driven or biased by a single SNP that might have a particularly large horizontal pleiotropic effect, we can re-estimate the effect by sequentially dropping one SNP at a time. Identifying SNPs that, when dropped, lead to a dramatic change in the estimate can be informative about the sensitivity of the estimate to outliers.

### Funnel plots

A tool used in meta-analysis is the funnel plot in which the estimate for a particular SNP is plotted against its precision (*Sterne et al., 2011*). Asymmetry in the funnel plot may be indicative of violations of the IV2 assumption through horizontal pleiotropy.

## Other MR analysis methods

In addition to the above, MR-Base also supports access to the following statistical methods implemented in other R packages: an extension of the IVW method that allows for correlated SNPs (*Yavorska and Burgess, 2017*), a method for the detection and correction of outliers in IVW linear regression (MR-PRESSO, *Verbanck et al., 2018*), methods for fitting and visualising radial IVW and radial MR-Egger models (*Bowden et al., 2017b*) and a method for correcting for horizontal pleiotropy using MR-RAPS (2SMR using robust adjusted profile scores (*Zhao et al., 2018*).

## Results

### The MR-Base database resource

We created a repository for complete GWAS summary data, where complete refers to all SNPs reported in a GWAS analysis with no exclusions according to p-values for the association with the trait of interest (e.g. datasets were not restricted to statistically significant SNPs). We included summary data from any array-based analysis, including targeted and untargeted arrays, with or without additional imputation for ungenotyped SNPs. The targeted arrays included immunochip and metabochip, as well as replication and fine-mapping studies with $\geq$10,000 variants. As of December 2017, the repository was populated by curated and harmonised datasets from 1673 GWAS analyses, corresponding to approximately 11 billion SNP-trait associations in 4 million samples (median sample size per study: 21,315). Excluding replication and fine-mapping studies, the median number of SNPs per study was 6.1 million (minimum = 79,129, maximum = 22,434,434); 95% of studies reported $\geq$393,465 SNPs. The current database also includes nine studies with $\leq$64,494 SNPs that generally correspond to replication and fine-mapping studies. The analysed traits included 605 traits

generated using the UK Biobank resource (*Millard et al., 2017*; *Bycroft et al., 2017*; *Churchhouse and Neale, 2017*;*GIANT consortium et al., 2016*; *Jones et al., 2016*; *Pilling et al., 2016*), 575 metabolomic traits from two studies (*Shin et al., 2014*; *Kettunen et al., 2016*), 151 immunological traits from one study (*Roederer et al., 2015*), and 342 other complex traits and diseases acquired from 123 GWAS publications (*Supplementary file 1A*). The latter publications corresponded to 79 studies, including 39 consortia and three cohorts. *Supplementary file 1A* provides a detailed overview of the available studies with complete summary data in MR-Base at the time of writing but the number is updated on a regular basis.

In addition to the 'complete summary data', we also collected published GWAS associations that comprise only the significant hits of a GWAS after applying stringent p-value thresholds (e.g. $p < 5 \times 10^{-8}$, a conventional threshold for declaring statistical significance in GWAS). These 'top hits', which can be used to define genetic instruments for exposures in MR analyses (see Materials and methods), include 29,792 SNPs obtained from clumping analysis of 1002 traits in the MR-Base database; 21,324 SNPs associated with 1628 complex traits and diseases in the NHGRI-EBI GWAS catalog (*Welter et al., 2014*); 187,318 SNPs associated with DNA methylation levels in whole blood at 33,256 genomic CpG sites, across five time points (*Gaunt et al., 2016*); 187,263 SNPs associated with gene expression levels at 27,094 gene identifiers, across 44 different tissues (*GTEx Consortium, 2015*); 1088 SNPs associated with metabolite levels in whole blood for 121 different metabolites (*Kettunen et al., 2016*); and 56 SNPs associated with protein levels in 47 different analytes (*Deming et al., 2016*).

The repositories of GWAS results described above can be interrogated and exploited for 2SMR using the R packages curated by MR-Base and other researchers. The R packages currently curated by MR-Base include 'TwoSampleMR' (https://github.com/MRCIEU/TwoSampleMR) and 'MRInstruments' https://github.com/MRCIEU/MRInstruments). Users can check the MR-Base website for updates to the curated packages. Accessible R packages curated by other researchers are described in *Supplementary file 1B*.

## Using MR-Base to estimate the causal relationship between LDL cholesterol and coronary heart disease

In an applied example, we conducted a MR study of the causal effect of LDL cholesterol on CHD risk, using summary data from the GLGC (*Willer et al., 2013*; *Do et al., 2013*) and CARDIoGRAM-plusC4D consortia (*Nikpay et al., 2015*), respectively. There were 91 studies (214,370 subjects) in the GLGC, 48 studies (195,813 subjects) in CARDIoGRAMplusC4D and 17 studies that were common to both consortia (including 59,970 subjects). We estimated that 31% of subjects in CARDIoGRAMplusC4D are also part of the GLGC and 28% of GLGC participants are also part of CARDIoGRAMplusC4D. The selected instruments (*Supplementary file 1C*) reportedly explained 2.4% of the variance in LDL cholesterol levels (*Willer et al., 2013*), equivalent to to an F statistic of 85 in the GLGC. This indicates that the instrument is strong and therefore unlikely to be susceptible to weak instrument bias or bias from sample overlap (*Burgess et al., 2011*).

The random effects IVW estimate indicated that the odds ratio (OR) (95% confidence interval [CI]) for CHD was 1.45 (1.30–1.62) per standard deviation increase in LDL cholesterol (*Figure 4*). There was, however, strong evidence for heterogeneity amongst SNPs (Cochran's Q value = 122.5, p=4.72e-07) and funnel plot asymmetry (*Figure 4a and d*), suggesting that at least some of the SNPs exhibit horizontal pleiotropy (a violation of the IV2 assumption, as shown in *Figure 1b*). There was evidence for a negative intercept (−0.013 [s.e.=0.005], p=0.020) and stronger odds ratio (1.85 [1.48–2.32]) in MR-Egger regression (*Figure 4b*) indicating some amount of directional horizontal pleiotropy. Similar results to the IVW estimate were provided by the weighted median (1.56 [1.43-1.70]) and weighted mode (1.68 [1.56-1.80]) estimators (*Figure 4*). In a leave-one-out analysis, we sequentially excluded one instrument (SNP) at a time to assess the sensitivity of the results to individual variants, finding that no single instrument was strongly driving the overall effect of LDL cholesterol on CHD (*Figure 4c*).

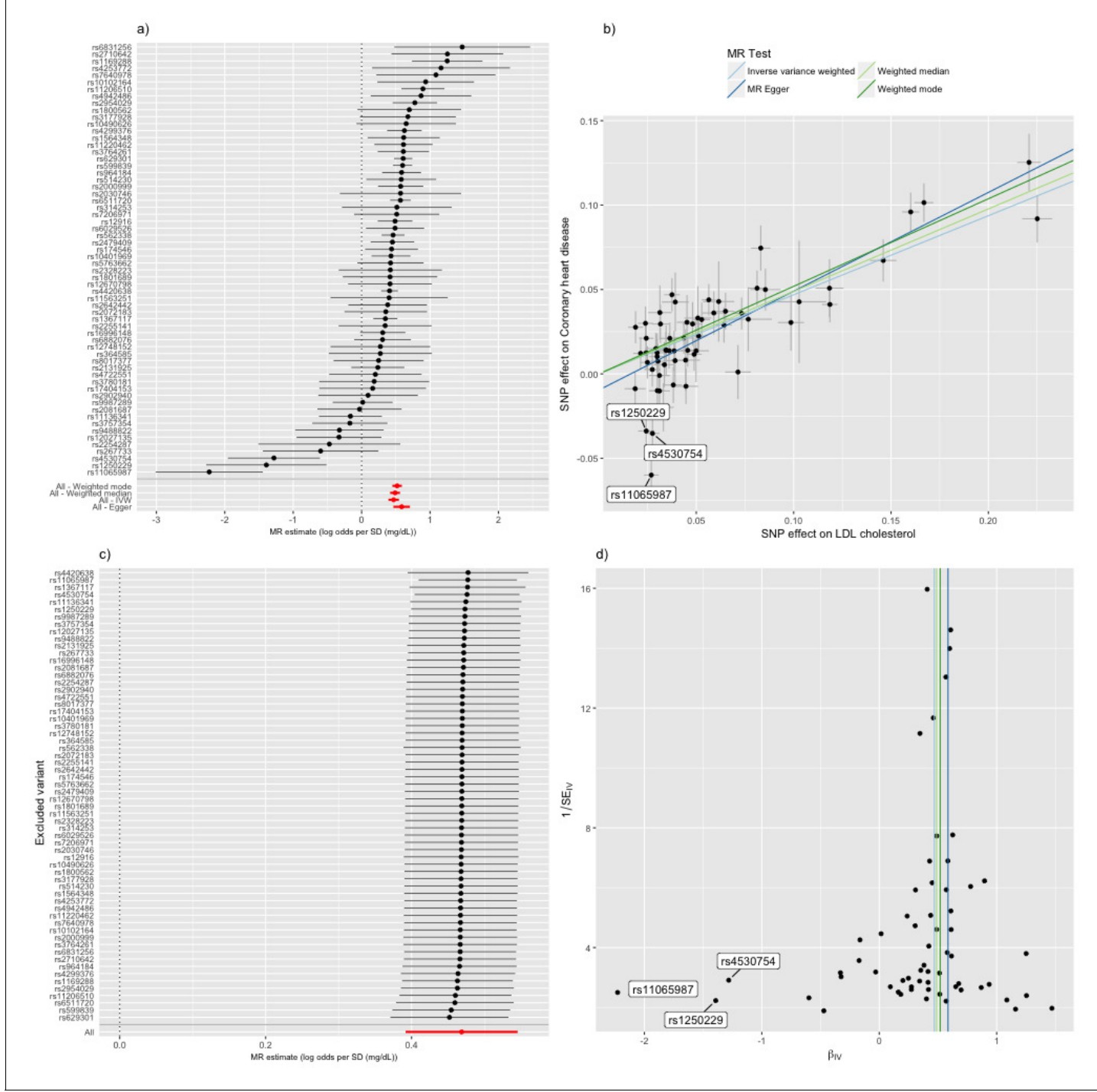

**Figure 4.** Mendelian randomization study of the effect of low density lipoprotein cholesterol levels on coronary heart disease. (**a**) A forest plot, where each black point represents the log odds ratio (OR) for coronary heart disease (CHD) per standard deviation (SD) increase in low density lipoprotein (LDL) cholesterol, produced using each of the 'LDL single nucleotide polymorphisms (SNPs)' as separate instruments, and red points showing the combined causal estimate using all SNPs together in a single instrument, using each of four different methods (weighted median, weighted mode, inverse-variance weighted [IVW] random effects and MR-Egger). Horizontal lines denote 95% confidence intervals. (**b**) A plot relating the effect sizes of the SNP-LDL association (x-axis, SD units) and the SNP-CHD associations (y-axis, log OR) with standard error bars. The slopes of the lines correspond to causal estimates using each of the four different methods. Outlier SNPs are labeled. (**c**) Leave-one-out sensitivity analysis. Each black point represents the IVW MR method applied to estimate the causal effect of LDL on CHD excluding that particular variant from the analysis. The red point depicts the IVW estimate using all SNPs. There are no instances where the exclusion of one particular SNP leads to dramatic changes in the overall result. (**d**) Funnel plot showing the relationship between the causal effect of LDL on CHD estimated using each individual SNP as a separate instrument against

*Figure 4 continued on next page*

Figure 4 continued

the inverse of the standard error of the causal estimate. Vertical lines show the causal estimates using all SNPs combined into a single instrument for each of four different methods. There is some asymmetry in the plot (an excess of strong protective effects associated with higher LDL cholesterol), which is potentially indicative of violations of instrumental variable (IV) assumptions, e.g. violation of the IV2 assumption through horizontal pleiotropy. Outlier SNPs are labeled.

DOI: https://doi.org/10.7554/eLife.34408.006

## Using PheWAS to interpret outliers and identify potential sources of horizontal pleiotropy

Inspection of the forest, funnel and scatter plots highlighted three outlier SNPs (rs11065987, rs1250229 and rs4530754) as potential sources of heterogeneity and the negative intercept in MR-Egger regression (*Figure 4*). For these three SNPs, the LDL-raising variant was associated with lower CHD risk, contrary to the results for the majority of the other LDL-raising variants. In such situations it is strongly advised to check for effect allele coding errors (*Hartwig et al., 2016*). We confirmed that the SNPs were not palindromic (such SNPs are particularly prone to coding errors in 2SMR) and that the LDL and CHD risk variants were compatible with those reported in the GWAS catalog and the original study reports. The unusually strong cardio-protective effects of the three LDL raising variants are also compatible with horizontal pleiotropy, whereby the effects of the SNPs on CHD are independent of their effects on LDL cholesterol.

To identify potential sources of horizontal pleiotropy, we performed a PheWAS of rs11065987, rs1250229 and rs4530754, using a threshold of $p<2.04e-05$ (0.05/2453 'trait lookups') to select traits for further evaluation. Only rs11065987, located upstream of the *BRAP* gene, was associated with non-lipid non-vascular-disease traits, including two markers of adiposity (body mass index and hip circumference); three blood pressure traits (diastolic blood pressure, systolic blood pressure and mean arterial pressure); five hematological markers (hematocrit, haemoglobin concentration, packed cell volume, red blood cell count and platelet count); five autoimmune diseases (inflammatory bowel disease, primary biliary cirrhosis, Crohn's disease, rheumatoid arthritis and celiac disease); four metabolites (urate, kynurenine, erythronate and C-glycosyltryptophan); serum cystatin C (a marker of kidney function); and tetralogy of Fallot (*Supplementary file 1D*).

In further MR analyses of these traits, we found that higher hematocrit, higher blood pressure, higher BMI and higher hip circumference were putatively associated with higher CHD risk ($p<0.05$) (*Supplementary file 1D*). However, of these traits, only the indirect effect of rs11065987 due to hematocrit (-0.012, SE=0.005) was in the same direction as the direct effect of rs11065987 on CHD ($-0.060$, SE = 0.011), whereas the indirect effects mediated by hip circumference (0.004, SE = 0.002), BMI (0.007, SE = 0.001) and LDL cholesterol (0.011, SE = 0.001) were in opposite directions. Due to unreported effect alleles in the relevant GWAS, we were unable to assess the indirect effect of rs11065987 mediated by blood pressure. These results suggest that at least one cardio-protective mechanism for the LDL-raising variant of rs11065987 is due to a pleiotropic effect of hematocrit. However, this result should be interpreted with caution and requires replication in independent studies and further examination for potential violations of MR assumptions in sensitivity analyses.

## Performing hypothesis-free searches for causal relationships

In order to gain insight into potential opportunities for repurposing or adverse effects of LDL cholesterol lowering - an established intervention stategy for CHD prevention - we conducted a hypothesis-free MR-PheWAS analysis (*Millard et al., 2015*; *Haycock et al., 2017*). Instrumented using 62 SNPs (*Supplementary file 1C*), we obtained fixed effects IVW estimates of lower LDL cholesterol on 40 non-vascular diseases and 108 non-lipid complex traits in MR-Base (*Figure 5*). Using an unadjusted p-value of 0.05 to denote suggestive evidence for association, we identified 16 non-vascular traits associated with LDL cholesterol. Surpassing a 5% false discovery threshold were lower mortality measures, higher adiposity measures, and higher risk of type two diabetes. We emphasise that this analysis is shown here for purposes of demonstrating the utility of MR-Base for efficiently screening many traits for hypothesis generation, and any claims of causality must be followed up with rigorous examination of potential violations of MR assumptions in sensitivity analyses, replication in

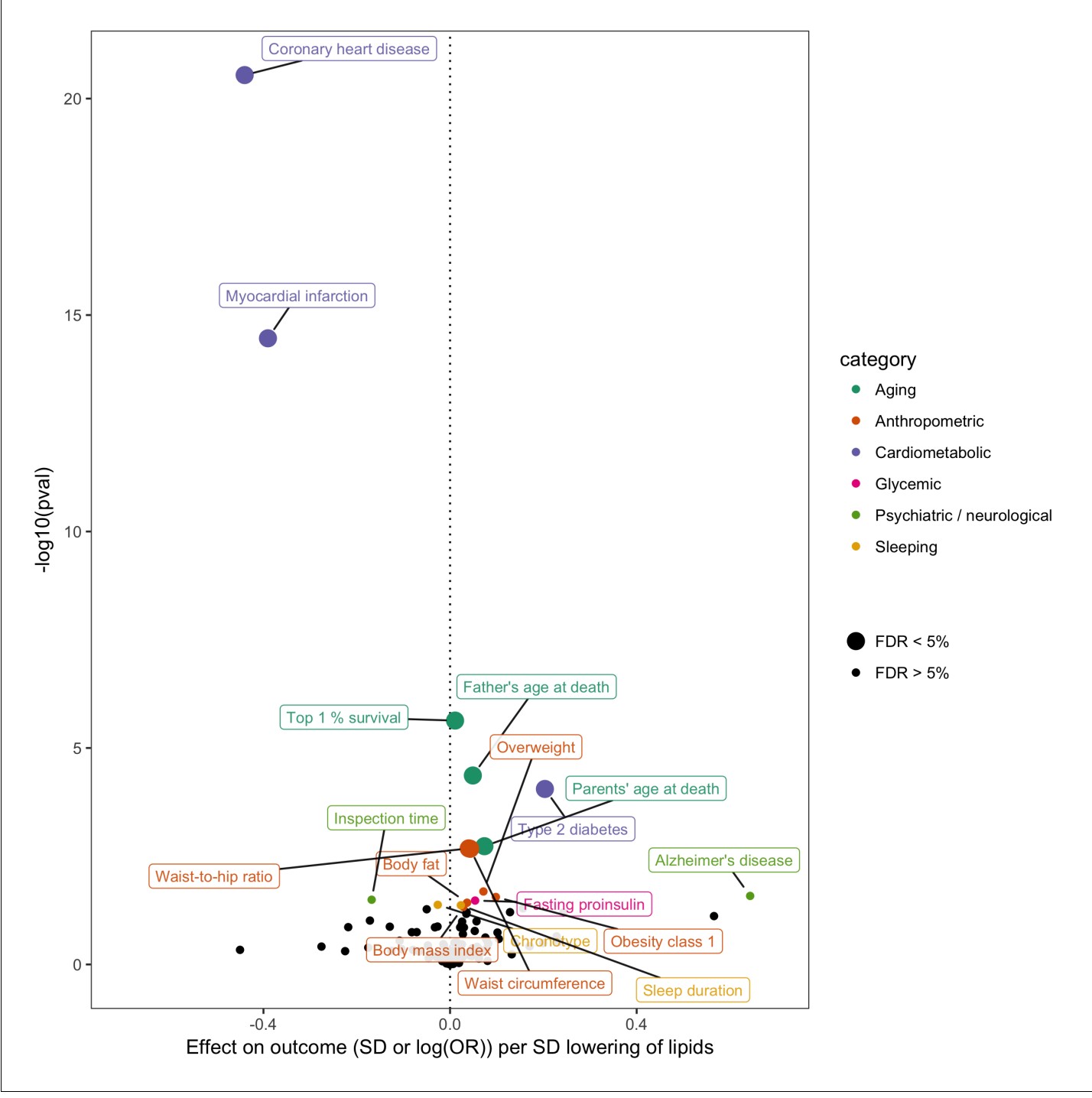

**Figure 5.** Effect of lower low density lipoprotein cholesterol on 150 traits in MR-Base. The x-axis shows the standard deviation (SD) change or log odds ratio (OR) for each of 150 traits per SD decrease in low density lipoprotein (LDL) cholesterol. The y-axis shows the p-value for the association on a -log10 scale. The effects on the x-axis correspond to the slope from fixed effects inverse variance weighted (IVW) linear regression of single nucleotide polymorphism (SNP)-outcome effects regressed on the SNP-LDL effects. Those results that have a p-value<0.05 are labelled. Larger points denote false discovery rate (FDR) < 0.05. LDL cholesterol was instrumented by 62 SNPs.

DOI: https://doi.org/10.7554/eLife.34408.007

independent studies and triangulation with evidence from other study designs (*Lawlor et al., 2016*; *Munafò and Davey Smith, 2018*).

## Discussion

As the availability of published GWAS summary data continues to grow, MR offers an attractive approach for exploring the aetiology of disease (*Pickrell et al., 2016*). We have developed a platform that integrates a database of GWAS summary data together with a public repository of statistical methods for enabling systematic causal inference across the phenome. This benefits modelling of phenotypic relationships in three ways. First, it greatly simplifies and expedites the practice of performing MR and PheWAS analyses. Second, automating the application of state-of-the-art methodology establishes basic standards for reporting MR results and improves the reliability and reproducibility of causal inference. Third, it maximises the breadth of possible causal relationships that can be interrogated by drawing together genetic information on as many traits as possible. This presents new analytical opportunities that may not have been feasible previously.

In an applied example, we used MR-Base to recapitulate the known (*Holmes et al., 2017*) causal effect of higher LDL cholesterol on CHD risk, but also found strong evidence for violations of assumptions. The latter was at least partly explained by three LDL raising variants that were associated with lower CHD risk. One of the variants, rs11065987, was subject to substantial pleiotropy, suggesting that its association with CHD may operate independently of its effect on LDL cholesterol, with hematocrit identified as a potential alternative pathway. Despite the evidence for violations of IV assumptions, results remained compatible with a causal effect of higher LDL cholesterol on CHD in sensitivity analyses and were similar to findings from clinical trials (*Silverman et al., 2016*; *White et al., 2016*) and observational studies (*Di Angelantonio et al., 2009*). We also showed putative evidence that lower LDL cholesterol is causally related to lower mortality, but higher adiposity and higher risk of type two diabetes. The latter result is compatible with trials that have shown that lipid lowering drugs similarly increase risk of type two diabetes (*Swerdlow et al., 2015*; *Schmidt et al., 2017*; *Sattar et al., 2010*). These results were shown to demonstrate the utility of MR-Base for hypothesis-free MR-PheWAS analyses. We refrain from making strong claims of causality here, but the latter results highlight the scope of MR-Base for identifying opportunities for drug repurposing or potential adverse effects of pharmaceutical and public health interventions.

Much has been written about the strengths and limitations of MR (*Davey Smith and Hemani, 2014*; *Haycock et al., 2016*; *Swerdlow et al., 2016*; *Holmes et al., 2017*; *VanderWeele et al., 2014*; *Lawlor et al., 2008*; *Lawlor et al., 2016*; *Zheng et al., 2017a*), and we summarise how these relate to MR-Base in *Supplementary file 1E*, with particular focus on the most important strengths and limitations below.

### Strengths of MR-Base

#### Efficiency of performing 2SMR

Automation of 2SMR greatly facilitates the practical implementation of the analysis, from acquiring, checking and harmonising GWAS summary data, and identifying LD proxies where necessary, to performing a battery of inferential and sensitivity analyses. In addition to the tests that we have implemented, we have also made it straightforward to apply other R packages that implement MR methodology to the MR-Base database. We advocate that causal inference is presented in a triangulation framework, consolidating evidence from different study designs (*Lawlor et al., 2016*; *Munafò and Davey Smith, 2018*). With automation of 2SMR, researchers are afforded more bandwidth to integrate evidence from other, complementary, study designs.

#### Reliability of MR estimates

MR-Base seeks to improve reliability in two ways. First, automation of data handling procedures with a rigorous quality control (QC) pipeline can eliminate errors that might arise due to human error. It has been shown that discrepant results can easily arise using identical data because of potential errors of implementation (*Hartwig et al., 2016*; *Hartwig et al., 2017*; *Inoshita et al., 2018*). Although MR-Base uses a number of strategies to reduce the potential for effect allele coding errors (see Materials and methods), we strongly encourage users to check that effect signs (i.e. positive or

negative betas) accurately reflect the correct effect allele, in both the outcome and exposure study. Second, bias can arise in MR analyses if its assumptions are not met, and some of these are unprovable. We enable users to easily perform state-of-the-art methods and sensitivity analyses to evaluate their results, and the repository of methods available to users will continually expand as new ones become available.

## Reproducibility of 2SMR results
The integration of data and analytical software under a single platform means that the results presented by researchers can easily be reproduced by other analysts. Diagnostic reports are automatically generated, and if using the web application we generate R code that can be used to reproduce any results that are derived through the graphical user interface.

## A central data repository of complete GWAS summary data
Through our PheWAS of MR outliers we have shown the utility of having GWAS summary data together in one location. We have developed a highly scalable database infrastructure which can be used to store new and existing studies, and that can be easily and rapidly queried through a web app or an application programming interface (API). Researchers may wish to only make their GWAS summary data publicly available after a certain time period; to this end, we have developed an authentication system that allows fine-grained permission controls to restrict access for particular datasets to specific users, where necessary.

## Avoiding reverse causal instruments
MR-Base provides the Steiger test, a method for identifying the correct direction of effect (*Hemani et al., 2017b*). The use of p-value thresholds for instrument selection can increase the potential for 'reverse causal' instruments, which occurs when the effect of genetic variants on a hypothesized exposure actually occurs via the hypothesized outcome, and can lead to incorrect inferences about directions of causality (*Hemani et al., 2017a*). One way to mitigate the impact of such invalid instruments is to check that the selected genetic variants are more strongly associated with the hypothesized exposure than with the outcome. It is important to note, however, that this test is subject to a number of assumptions (*Hemani et al., 2017b*).

## **Limitations that are inherent to 2SMR**
### Sample overlap
GWAS typically involves meta-analyses of large numbers of different cohorts, where each cohort is likely to contribute to GWAS on many different traits. As a consequence there is likely to be considerable sample overlap amongst different GWAS within the database (e.g. we estimated that 28% of GLGC participants were also part of the CARDIoGRAMplusC4D consortium), which could bias effect estimates from 2SMR towards the confounded observational association - an example of weak instrument bias (*Pierce and Burgess, 2013*). Bias from sample overlap is made worse if some or all of the overlapping studies are also discovery studies for the SNP-exposure or SNP-outcome associations - due to the phenomenon of winner's curse (*Haycock et al., 2016*; *Bowden and Dudbridge, 2009*). We therefore strongly advise users of MR-Base to estimate the degree of participant overlap amongst the exposure, outcome and discovery GWAS in their analyses. Bias from sample overlap can be minimized by using strong instruments (e.g. F statistic much greater than 10 for the instrument-exposure association) (*Pierce and Burgess, 2013*). In our applied example, although we estimated some overlap between the GLGC and CARDIoGRAMplusC4D consortia, the robust strength of our instrument for LDL cholesterol (an F statistic of 85) likely minimized any potential impact of weak instruments bias in our results. We also encourage users to conduct sensitivity analyses using replication studies to define their instruments. See Pierce and Burgess (*Pierce and Burgess, 2013*) for further details on the relationship between instrument strength and bias from sample overlap.

### Samples derived from the same population
Although the exposure and outcome studies used in 2SMR should not involve overlapping participants, the participants from both studies should be from the same population - practically defined as being of similar age and sex distributions and geographic and ancestral origins (*Angrist and*

*Krueger, 1995*) - with similar patterns of LD in the genomic regions used to define the exposure. When exposure and outcome studies are derived from different populations, this could bias the magnitude of the association between the exposure and outcome. This could arise, for example, as a result of differences in LD between the SNPs used as instruments for the exposure and the underlying causal variants for the exposure, resulting in differences in instrument-exposure effect sizes (note that knowledge of causal variants is not required for MR). In addition, instrument-exposure effect sizes may vary with age or may differ between males and females. MR-Base provides meta-data on population characteristics, such as geographic origin, to help guide the user in selecting the most appropriate populations for their analysis. However, subtle population differences between studies of broadly similar backgrounds may persist. On the other hand, differences in population backgrounds should not generally increase the likelihood of incorrectly inferring an association when none exists (*Burgess et al., 2016*). Thus, results may still be informative for directions of causality, but not the magnitude of the effect, when the exposure and outcome studies are derived from different populations.

## Limitations compounded by MR-Base

### Multiple testing

Although automation allows more complex study designs, such as many-to-many MR studies and PheWAS, this also increases the scope for multiple testing, false positives and data dredging (whereby users do not present the results from all their analyses but cherry pick a few on the basis of arbitrary p-value thresholds). Users of MR-Base should adhere to well defined analysis plans and should be fully transparent about all traits investigated, including all exploratory analyses. Traditional MR studies, which are hypothesis-driven and done in conjunction with findings from observational studies, should be less prone to these biases.

### Interpretation

Automation shifts the problem of handling complex data to a problem of interpreting complex results. In a hypothesis-free study, involving many exposure-outcome combinations and multiple methods, a critical evaluation of each causal estimate may not be possible and could increase the potential for cherry picking of results. In such a situation, it may be more appropriate to consider a machine learning framework to select the most appropriate method and instrument selection strategy for each exposure-outcome analysis (*Hemani et al., 2017a*).

### Biased effects from automated instrument selection

Ideally, one would only use replicated GWAS effect sizes for instrumental variables because this reduces the impact of winner's curse (*Zollner and Pritchard, 2007*; *Bowden and Dudbridge, 2009*). In the presence of winner's curse the causal effect estimates are liable to be biased towards the null, assuming the exposure and outcome studies are independent (in the presence of substantial overlap between exposure and outcome studies winner's curse can compound the effect of weak instruments bias). The MR-Base database of complete summary data is currently biased towards discovery GWAS, with few results from replication studies. This means that the clumping procedure used to obtain independent significant hits will likely return instruments with inflated effect sizes. Similar problems exist for the 'omic QTL datasets, which seldom have independent replication data available. We advise users to assess the sensitivity of their results to winner's curse, e.g. using independent studies to redefine their instruments. MR-Base currently includes GWAS results for 605 traits from UK Biobank that could be used for this purpose and this number will increase over time.

## General advice for interpretation of results

When the biases and limitations indicated above are avoidable, users should consider modifying their analyses. For example, users could use an automated approach for instrument selection in their primary analysis but use manually curated instruments in sensitivity analyses of prioritised results. Sometimes, however, biases may be unavoidable, in which case users should acknowledge their possible impact and relax their conclusions. For example, inferences about directions of causality, shared genetic architecture (*Burgess et al., 2014*) or null effects (*VanderWeele et al., 2014*) are

usually much more reliable than inferences about magnitudes of causal effects, which are very sensitive to violations of assumptions.

## Related resources

A number of resources are available for MR analysis or extracting and using GWAS summary data. The MendelianRandomization R package (*Yavorska and Burgess, 2017*) is a standalone tool comprising several 2SMR methods and, in addition to the methods that we have implemented, we make it easy to import data from MR-Base into that R package. PhenoScanner (*Staley et al., 2016*) expands upon established GWAS catalogs (*Welter et al., 2014*; *Li et al., 2016*) by storing a large number of complete summary level datasets and providing a web interface for specific SNP lookups. SMR (*Zhu et al., 2016*) has been developed to automate colocalisation analysis between eQTLs and complex traits.

## Summary

There has been a massive growth in the phenotypic coverage and statistical power of GWAS over the past decade (*Visscher et al., 2017*; *Welter et al., 2014*). Many approaches to studying complex traits and diseases can now be interrogated using GWAS summary level data. By harvesting and harmonising these data into a database and directly integrating this with analytical software for 2SMR, MR-Base greatly improves the efficiency and reliability of hypothesis-driven approaches. The database is a generic repository of GWAS summary data accessible via an API and future work will see extensions to support other methods for the investigation of complex trait genetic architecture, such as fine-mapping (*Benner et al., 2016*), colocalisation (*Zhu et al., 2016*; *Newcombe et al., 2016*; *Giambartolomei et al., 2014*) and polygenic risk prediction (*Dudbridge, 2013*; *Euesden et al., 2015*). The curation of data and methods achieved by MR-Base opens up new opportunities for hypothesis-free and phenome-wide approaches.

## Resources

The following resources are all part of the MR-Base platform:

- Website: http://www.mrbase.org/
- MR-Base web application: http://app.mrbase.org/
- MR-Base PheWAS application: http://phewas.mrbase.org/
- TwoSampleMR R package: https://github.com/MRCIEU/TwoSampleMR
- TwoSampleMR documentation: https://mrcieu.github.io/TwoSampleMR/
- MRInstruments R package: https://github.com/MRCIEU/MRInstruments
- MR-Base database API: http://api.mrbase.org/

Code to reproduce the analysis in this paper: https://github.com/explodecomputer/mr-base-methods-paper.

# Materials and methods

## Overview

MR-Base comprises two main components: a database of GWAS summary association statistics and LD proxy information, and R packages and web applications for causal inference methods. The GWAS summary data are further structured into complete summary data (i.e. all SNP-phenotype associations) and 'top hits', which comprises subsets of GWAS summary data (typically the statistically significant results). The R packages include the TwoSampleMR (https://github.com/MRCIEU/TwoSampleMR) package, which supports data extraction, data harmonisation and MR analysis methods, and the MRInstruments (https://github.com/MRCIEU/MRInstruments) package, which is a repository for instruments. MR-Base also supports access to R packages curated by other researchers, including MendelianRandomization (*Yavorska and Burgess, 2017*) (https://cran.r-project.org/web/packages/MendelianRandomization/), RadialMR (*Bowden et al., 2017b*) (https://github.com/wspiller/radialmr), MR-PRESSO (*Verbanck et al., 2018*) (https://github.com/rondolab/MR-PRESSO) and mr.raps (*Zhao et al., 2018*) (https://github.com/qingyuanzhao/mr.raps). The available methods are updated on a regular basis. The database is accessible through an API (http://api.mrbase.org)

and is therefore extendable to other causal inference methods not currently covered by the afore-mentioned packages. A web app (http://app.mrbase.org) was developed as a user-friendly wrapper to the R package using the R/shiny framework. The SNP lookup tool is available through the PheWAS web app (http://phewas.mrbase.org/). Scripts to perform the analyses presented in this paper are available at https://github.com/explodecomputer/mr-base-methods-paper (*Hemani, 2018*; copy archived at https://github.com/elifesciences-publications/mr-base-methods-paper).

## MR-Base database

### Obtaining summary data from genomes wide association studies

We downloaded publicly available datasets from study-specific websites and dbGAP and invited studies curated by the NHGRI-EBI GWAS catalog to share results (if these were not already publically available). To be eligible for inclusion in MR-Base, studies should provide the following information for each SNP: the beta coefficient and standard error from a regression model (typically an additive model) and the modelled effect and non-effect alleles. This is the minimum information required for implementation of 2SMR. The following information is also sought but is not essential: effect allele frequency, sample size, p-values for SNP-phenotype associations, p-values for Hardy–Weinberg equilibrium, p-values for Cochran's Q test for between study heterogeneity (if a GWAS meta-analysis) and metrics of imputation quality, such as info or $r^2$ scores (for imputed SNPs). MR-Base also includes information on the following study-level characteristics: sample size, number of cases and controls (if a case-control study), standard deviation of the sample mean for continuously distributed traits, geographic origin and whether the GWAS was conducted in males or females (or both). In future extensions to MR-Base, we plan to collate more detailed information on phenotype distributions (e.g. sample means for continuously distributed phenotypes) and population character-istics (mean and standard deviation of age and number of males and females) and statistical models (e.g. covariates included in regression models and genomic control inflation factors).

## Creating a database of harmonized GWAS summary data

GWAS summary data posted online tends not to follow standardised formats, therefore harmonisa-tion of these disparate data sources is a manual process. We adopted a systematic approach to har-monize these data and developed an Elasticsearch database (https://www.elastic.co/products/elasticsearch, v5.6.2) to store, structure and query the harmonised data. Insofar as it was possible we recorded all QC and harmonisation processes for each of the 1673 datasets to aid with reproducibil-ity. The following steps were taken for each dataset:

### Step 1. Pre-cleaning data

Prior to file harmonisation we carried out the following pre-cleaning steps: dropped duplicate data-sets, removed non–SNP-level meta-information, removed unexpected characters such as non-utf8 characters, split 95% confidence intervals contained within a single column into two columns. In addition, when results from a single GWAS were split across separate chromosome files, we com-bined these into a single file. When a single file contained results for multiple traits, we split the file into separate files for each trait.

### Step 2. Collecting and arranging meta-data

In order to collect the key information and harmonize these data into a uniform format, the column headers of each of the pre-cleaned GWAS files were collected using bash shell scripts and stored in Google sheets. The column headers were then reordered and columns with required information (such as file path, file name, file suffix, SNP rs ID, effect allele, other allele, effect size, standard error, p-value and sample size) were recorded.

### Step 3. Converting summary data into a uniform format

After collecting and rearranging the column information, we automatically converted the files into uniform formats, converting odds ratios to log odds ratios, estimating standard errors from 95% con-fidence intervals or, when the latter were missing, from effect estimates and p-values, and inferring sample size from overall sample size if SNP-level sample size was missing. The output files are tab-delimited and contain 8 columns of information: SNP rs number, effect allele, other allele, effect

allele frequency, beta (effect estimate for continuous traits or log odds ratio for binary traits), standard error, p-value and sample size.

## Step 4. Controlling quality of data

In order to control the quality of the data we: drop duplicate SNP records, remove records with missing SNP ID or p-value, check mislabelled columns, check data type for each cell of data (i.e. data should be ASCII format for the SNP ID column; string (A, G, C or T) for effect allele and other allele column; numerical value between 0 and 1 for effect allele frequency column and p-value column; numerical value for beta column; numerical value larger than 0 for standard error column; and a numerical value larger than 0 for sample size column). The identity of the effect allele column is confirmed by checking the meta-data or readme files accompanying the GWAS results, correspondence with the original study or comparison with the NHGRI-EBI GWAS catalog.

## Step 5. Building the relationship of data and creating an elasticsearch index

The cleaned and harmonized data were then indexed using Elasticsearch (https://www.elastic.co/products/elasticsearch, v5.6.2) to create a search engine. As shown in *Supplementary file 1G*, the database is structured into sets of indices (datasets) for association data, with two MySQL tables for study and LD proxy data:

- The study table (MySQL) contains study-level information, including file name, trait name, study name (or first author of relevant GWAS publication if study name missing), geographic origins of the study, whether the study contains males or females (or both), number of cases, number of controls, sample size, PubMed identifier (PMID), publication year, units of the SNP-trait effect (e.g. mg/dL, log odds, etc); standard deviation of the sample mean for the trait (if continuously distributed); manually curated trait category (e.g. disease or risk factor); and manually curated trait subcategory (e.g. cardiovascular disease, cancer, anthropometric risk factor, etc).
- The association datasets (Elasticsearch) contain the SNP association information including study ID, SNP ID, effect allele, other allele, beta, standard error, p-value and sample size. Effect allele refers to the modelled or coded allele in linear or logistic regression (typically using an additive genetic model); other allele refers to the non-effect allele; the beta, standard error and p-value refer to the change in trait per copy of the effect allele. Separate indices exist for major 'batches' of data, but are queried in combination by the API, requiring no additional operations on the part of the user.
- The 'SNP proxies' table (MySQL) provides a list of proxy SNPs for when the test SNP is missing from the requested study and increases the chance of identifying the SNP in both the exposure and outcome data.

## Linkage disequilibrium proxy data

One of the main functions of the MR-Base database is to provide association data for requested SNPs from studies of interest to the user (*Figure 2*). Often, however, a requested SNP may not be present in the requested GWAS (e.g. because of different imputation panels or because imputed SNPs were not available). In order to enable information to be obtained even when SNPs are missing, we provide an LD proxy function using 1000 genomes data from 503 European samples. For each common variant (minor allele frequency [MAF]>0.01) we used plink1.90 beta three software to identify a list of LD proxies. We recorded the $r^2$ values for each LD proxy and the phase of the alleles of the target and proxy SNPs. We limited the LD proxies to be within 250 kb or 1000 SNPs and with a minimum $r^2 = 0.6$.

## MR instrument catalogs

We have assembled a collection of strong SNP-phenotype associations from various sources that can be used as potential instruments in Mendelian randomization studies. Instruments are currently restricted to biallelic SNPs but in principle could be extended in future versions to accommodate multi-allelic SNPs or copy number variants (CNVs). The potential instruments generally correspond to the 'top hits' from a GWAS, rather than the entire collection of GWAS summary statistics. As such, the traits included here can only be evaluated as potential exposures in a hypothesized exposure-outcome analysis (complete summary data are required when evaluating traits as potential

outcomes). All curated instruments are available through the MRInstruments R package (https://github.com/MRCIEU/MRInstruments).

### NHGRI-EBI GWAS catalog

This is a comprehensive catalog of reported associations from published GWAS (*Welter et al., 2014*). To make the data suitable for Mendelian randomization, we converted odds ratios to log odds ratios and inferred standard errors from reported 95% confidence intervals or (if the latter were unavailable) from the reported p-value using the Z distribution. The GWAS catalog scales odds ratios to be greater than 1 and includes information on unit changes (e.g. mg/dl increase) for beta coefficients. In MR-Base we therefore assume that effect sizes are odds ratios if they are greater than 1 and are missing information on unit changes. We extracted information on the units of the SNP-trait effect; and identified effect and non-effect alleles by comparing the risk allele reported in the GWAS catalog to allele information downloaded from ENSEMBL, using the R/biomaRt package (*Durinck et al., 2009*). R/biomaRt was also used to identify base pair positions (in GRCh38 format) and associated candidate genes. We inferred effect allele frequency from the risk allele frequency reported in the GWAS catalog. We excluded SNP-trait associations from the GWAS catalog if they were missing a p-value, beta (estimate of the SNP-trait effect) or a standard error for the beta. The MR-Base standardized version of the GWAS catalog (2017-03-20 release at the time of writing) comprises 21,324 potential instruments for 1628 traits. There are, however, several caveats to using the GWAS catalog as a source of instruments. First, reported units of analysis are often unclear (e.g. results are often reported as reflecting a 'unit increase'). Second, the GWAS catalog prioritises results from the largest reported analysis in the original study report (typically the discovery study or a meta-analysis of discovery and replication studies). This makes instruments from the GWAS catalog susceptible to winner's curse, which can compound the effect of weak instruments bias.

### Accessible resource for integrated epigenomics studies (ARIES) mQTL catalog

We obtained a large set of SNPs associated with DNA methylation levels (i.e. mQTLs) using the ARIES dataset (*Gaunt et al., 2016*). mQTLs were identified in 1000 mothers at two time points and 1000 children at three time points. Top hits were obtained from http://mqtldb.org with p<1e-7. There are 33,256 unique CpG sites across the five time points with at least one independent mQTL. These mQTLs can be used as instruments for DNA methylation at CpG sites in Mendelian randomization analyses. The mQTLs can also be used to perform methylome-wide association studies (MWAS), to evaluate the association between DNA methylation at each CpG site and a phenotype of interest (implementable in MR-Base through the Wald ratio method when only a single mQTL for a CpG site is available).

### GTEx eQTL catalog

We used the GTEx resource (*GTEx Consortium, 2015*) of published independent cis-acting expression QTLs (cis-eQTLs) to create a catalog of SNPs influencing up to 27,094 unique gene identifiers across 44 tissues. These eQTLs can be used as instruments for gene-expression in Mendelian randomization analyses. The eQTLs can also be used to perform transcriptome-wide association studies (TWAS), to evaluate the association between expression of each gene and a phenotype of interest (implementable in MR-Base using the Wald ratio method when only a single eQTL for a gene is available).

### Metabolomic QTL catalog

SNPs influencing 121 metabolites measured using nuclear magnetic resonance (NMR) analysis in whole blood were obtained (*Kettunen et al., 2016*), totalling 1088 independent QTLs across all metabolites.

### Proteomic QTL catalog

SNPs influencing 47 protein analyte levels (*Deming et al., 2016*) in whole blood were obtained, totaling 57 independent proteomic QTLs.

## Defining instruments using the MR-Base catalog of complete summary data

The MR-Base repository of complete GWAS summary data, which contains all SNPs from a GWAS regardless of p-value, can also be used to define instruments. This involves extracting independent sets of SNP-phenotype associations that surpass user-specified p-value and clumping thresholds. However, as the MR-Base repository of complete summary data is based mostly on discovery studies, this strategy may be susceptible to false positive instruments (where some of the selected genetic variants are not truly associated with the target exposure) and winner's curse. See discussion and *Supplementary file 1E* for potential implications on results of these limitations.

### Interface to the data

The Elasticsearch database is behind a firewall and cannot be queried directly, in order to prevent misuse and to keep non-public data secure. An API (http://api.mrbase.org) is used to interface with the database, controlling access based on user permissions and using Google OAuth2.0 for user authentication.

A user friendly interface to the API is provided through the TwoSampleMR R package. A complete guide to use the R package is available at https://mrcieu.github.io/TwoSampleMR/ and a list of the analytical functions that are currently implemented are shown in *Supplementary file 1B*.

### Applied example

We used MR-Base to recapitulate the known (*Holmes et al., 2017*) causal effect of higher LDL cholesterol on CHD risk. To obtain the list of instruments for LDL cholesterol we searched for the GLGC entry (*Willer et al., 2013*) in the GWAS catalog dataset in the MRInstruments library. This returned 62 SNPs. Due to unclear effect size units in the GWAS catalog, we extracted these 62 SNPs from the MR-Base database to obtain effect sizes in standard deviation units. We then searched for these 62 SNPs in the CARDIoGRAMplusC4D GWAS dataset (*Nikpay et al., 2015*) in the MR-Base database. One of the SNPs was not available and an LD proxy was identified. We harmonised the dataset, setting the algorithm to assume all SNPs were coded with alleles on the forward strand. Disabling this option would have excluded 7 palindromic SNPs with allele frequencies close to 0.5.

The code to reproduce this analysis is below.

```
library(TwoSampleMR)
library(MRInstruments) data(gwas_catalog)
library(MRInstruments) data(gwas_catalog)
# Get published SNPs for LDL cholesterol
ldl_snps <- subset(gwas_catalog, grepl("LDL choles", Phenotype) & Author == "Wil-
ler CJ")$SNP
# Extract from GLGC dataset
exposure <- convert_outcome_to_exposure(extract_outcome_data(ldl_snps, 300))
# Get outcome data from Cardiogram 2015
outcome <- extract_outcome_data(exposure$SNP, 7)
# Harmonise exposure and outcome datasets
# Assume alleles are on the forward strand
dat <- harmonise_data(exposure, outcome, action=1)
# Perform MR
mr(dat)
mr_heterogeneity(dat)
# Label outliers and create plots
dat$labels <- dat$SNP dat$labels[! dat$SNP
%in% c("rs11065987", "rs1250229", "rs4530754")] <- NA
mr_plots(dat)
```

A PheWAS of outliers in the LDL-CHD MR results was conducted to identify potential sources of horizontal pleiotropy. First, we searched the MR-Base database of complete summary data and the GWAS catalog for associations with outlier SNPs, using a threshold of p<2.04e-05 (0.05/2453 'trait

lookups') to select traits for further evaluation. When identical traits were duplicated across different GWAS analyses, we retained the GWAS with the largest sample size. Of the outlier SNPs analysed, only rs11065987 was associated with non-lipid-non-vascular-disease traits and was therefore the only SNP retained for further analyses. We then conducted MR analyses to estimate the effect of the selected traits on CHD, scaled to reflect the magnitude of the observed effect of rs11065987 on the trait. Instruments were based on SNPs associated with the traits at a p-value less than 5e-8, with clumping to ensure independence between SNPs (clumping $r^2$ cutoff=0.001 and clumping window=10,000kb). The GWAS catalog was used to define instruments

when the selected trait was unavailable in the MR-Base database of complete summary data. All instruments excluded rs11065987. The effect of the traits on CHD was based on the slope from IVW linear regression, except where only a single SNP was available to instrument the trait, in which case the Wald ratio method was used. The variance of the slope from IVW linear regression was estimated using a random effects model, except where there was underdispersion in the causal estimates between SNPs, in which case a fixed effects model was used. We then compared the rs11065987-CHD effect (the direct effect) with the trait-CHD effect (indirect effect of rs11065987). Where effects were in opposite directions we concluded that it was less likely that the association between rs11065987 and CHD was mediated by the selected trait.

Complete code for all analyses can be found here: https://github.com/explodecomputer/mr-base-methods-paper (*Hemani, 2018*; copy archived at https://github.com/elifesciences-publications/mr-base-methods-paper).

## Acknowledgements

We gratefully acknowledge our collaborators who shared summary data: Nicole Soranzo on behalf of the HaemGen consortium; David A van Heel on behalf of the celiac disease GWAS; Yukinori Okada on behalf of the C-reactive protein GWAS; GliomaScan; Clara S. Tang, Merce Garcia-Barcelo and Paul KH Tam on behalf of the Hirschsprung's disease GWAS; Kaya Kvarme Jacobsen on behalf of the migraine in bipolar disorder GWAS; Gregory T Jones and Matthew J Bown on behalf of the International Aneurysm Consortium; Omar Albagha and Stuart H. Ralston on behalf of the Paget's disease GWAS; Andre Franke, Annegret Fischer and David Ellinghaus on behalf of the sarcoidosis GWAS; Asta Försti, Hauke Thomsen and Stefano Landi on behalf of the thyroid cancer GWAS; Heather Cordell on behalf of the UK, Italian and Canadian-US primary biliary cirrhosis GWAS; Ani W Manichaikul and R Graham Barr on behalf of the percent emphysema GWAS; Jeffrey E Lee on behalf of the melanoma GWAS of the MDACC study.

We also gratefully acknowledge all studies and databases that have made GWAS summary data available (the investigators of these studies and databases did not participate in the analysis, writing or interpretation of this report): **ADIPOGen** (Adiponectin genetics consortium), **AMDGene** (Age-related Macular Degeneration Gene Consortium), **BioBank Japan Project**, **C4D** (Coronary Artery Disease Genetics Consortium), **CARDIoGRAM** (Coronary ARtery DIsease Genome wide Replication and Meta-analysis), **CKDGen** (Chronic Kidney Disease Genetics consortium), **CORNET** (The CORtisol NETwork), **dbGAP** (database of Genotypes and Phenotypes), **DCCT/EDIC** (Diabetes Control and Complications Trial/Epidemiology of Diabetes Intervention and Complications study cohort), **DIAGRAM** (DIAbetes Genetics Replication And Meta-analysis), **EAGLE** (EArly Genetics and Lifecourse Epidemiology Consortium), **EAGLE Eczema** (EArly Genetics and Lifecourse Epidemiology Eczema Consortium), **EGG** (Early Growth Genetics Consortium), **ENIGMA** (Enhancing Neuro Imaging Genetics through Meta Analysis), **GABRIEL** (A Multidisciplinary Study to Identify the Genetic and Environmental Causes of Asthma in the European Community), **GCAN** (Genetic Consortium for Anorexia Nervosa), **GEFOS** (GEnetic Factors for OSteoporosis Consortium), **GIANT** (Genetic Investigation of ANthropometric Traits), **GIS** (Genetics of Iron Status consortium), **GLGC** (Global Lipids Genetics Consortium), **GliomaScan** (cohort-based genome-wide association study of glioma), **GPC** (Genetics of Personality Consortium), **GUGC** (Global Urate and Gout consortium), **HaemGen** (haemotological and platelet traits genetics consortium), **HRgene** (Heart Rate consortium), **IAC** (the International Aneurysm Consortium), **ICBP** (International Consortium for Blood Pressure), **IGAP** (International Genomics of Alzheimer's Project), **IIBDGC** (International Inflammatory Bowel Disease Genetics Consortium), **ILCCO** (International Lung Cancer Consortium), **ImmunoBase** (resource focused on the genetics and genomics of immunologically related human diseases), **IMSGC**

(International Multiple Sclerosis Genetic Consortium), **ISGC** (International Stroke Genetics Consortium), **MAGIC** (Meta-Analyses of Glucose and Insulin-related traits Consortium), **MDACC** (MD Anderson Cancer Center), **MESA** (Multi-Ethnic Study of Atherosclerosis), **NHGRI-EBI GWAS catalog** (National Human Genome Research Institute and European Bioinformatics Institute Catalog of published genome-wide association studies), **PanScan** (Pancreatic Cancer Cohort Consortium), **PGC** (Psychiatric Genomics Consortium), **Project MinE** consortium, **ReproGen** (Reproductive ageing Genetics consortium), **SSGAC** (Social Science Genetics Association Consortium), **TAG** (Tobacco and Genetics Consortium) and **TRICL** (Transdisciplinary Research in Cancer of the Lung consortium). We gratefully acknowledge the assistance of Dr Johannes Kettunen and Dr Benjamin Neale.

Supported by Cancer Research UK grant C18281/A19169 (the Integrative Cancer Epidemiology Programme) and the Roy Castle Lung Cancer Foundation (2013/18/Relton). The Medical Research Council Integrative Epidemiology Unit is supported by grants MC_UU_12013/1, MC_UU_12013/2 and MC_UU_12013/8. PCH is supported by a Cancer Research UK Population Research Postdoctoral Fellowship (C52724/A20138). Jack Bowden is supported by a MRC Methodology Research Fellowship (grant MR/N501906/1). DME supported by the NHMRC APP1125200, APP1137714. GH is supported by Wellcome (208806/Z/17/Z).

## Additional information

### Funding

| Funder | Grant reference number | Author |
|---|---|---|
| Wellcome | 208806/Z/17/Z | Gibran Hemani |
| Cancer Research UK | C18281/A19169 | Benjamin Elsworth<br>Kaitlin H Wade<br>Vanessa Y Tan<br>Nicholas J Timpson<br>Caroline Relton<br>Richard M Martin<br>Tom R Gaunt<br>Philip C Haycock |
| GlaxoSmithKline | | Valeriia Haberland |
| Biogen | | Denis Baird |
| Medical Research Council | Methodology Research Fellowship, MR/N501906/1 | Jack Bowden |
| National Institute for Health Research | NIHR Bristol BRC | Nicholas J Timpson |
| Wellcome | | Nicholas J Timpson |
| Australian Research Council | | David M Evans |
| National Health and Medical Research Council | APP1125200 | David M Evans |
| National Health and Medical Research Council | APP1137714 | David M Evans |
| Cancer Research UK | Population Research Postdoctoral Fellowship, C52724/A20138 | Philip C Haycock |
| Roy Castle Lung Cancer Foundation | 2013/18/Relton | Philip C Haycock |

The funders had no role in study design, data collection and interpretation, or the decision to submit the work for publication.

### Author contributions

Gibran Hemani, Conceptualization, Resources, Data curation, Software, Formal analysis, Methodology, Writing—original draft, Writing—review and editing; Jie Zheng, Data curation, Methodology,

Writing—original draft; Benjamin Elsworth, Data curation, Software, Methodology; Kaitlin H Wade, Valeriia Haberland, Denis Baird, Ryan Langdon, Vanessa Y Tan, James Yarmolinsky, Data curation; Charles Laurin, Hashem A Shihab, Software; Stephen Burgess, Jack Bowden, Methodology; Nicholas J Timpson, David M Evans, Supervision; Caroline Relton, Richard M Martin, Conceptualization, Supervision; George Davey Smith, Conceptualization, Supervision, Methodology; Tom R Gaunt, Conceptualization, Software, Supervision, Methodology, Writing—original draft, Project administration, Writing—review and editing; Philip C Haycock, Conceptualization, Supervision, Data curation, Formal analysis, Methodology, Writing—original draft, Project administration, Writing—review and editing

### Author ORCIDs
Gibran Hemani http://orcid.org/0000-0003-0920-1055
Jie Zheng https://orcid.org/0000-0002-6623-6839
Kaitlin H Wade http://orcid.org/0000-0003-3362-6280
Denis Baird http://orcid.org/0000-0003-4600-6013
Stephen Burgess https://orcid.org/0000-0001-5365-8760
Vanessa Y Tan https://orcid.org/0000-0001-7938-127X
Caroline Relton http://orcid.org/0000-0003-2052-4840
Richard M Martin http://orcid.org/0000-0002-7992-7719
George Davey Smith http://orcid.org/0000-0002-1407-8314
Tom R Gaunt https://orcid.org/0000-0003-0924-3247
Philip C Haycock http://orcid.org/0000-0001-5001-3350

### Decision letter and Author response
Decision letter https://doi.org/10.7554/eLife.34408.011
Author response https://doi.org/10.7554/eLife.34408.012

---

## Additional files

### Supplementary files
• Supplementary file 1. (**A**) Genome-wide association studies with complete summary data in MR-Base as of December 2017.
(**B**) List of Mendelian randomization analysis methods.
(**C**) Genetic instruments for low density lipoprotein cholesterol.
(**D**) Phenome-wide association study of LDL-C raising cardio-protective variant.
(**E**) Limitations of Mendelian randomization and potential solutions.
(**F**) Glossary of terms.
(**G**) The schema of the MR-Base database.
DOI: https://doi.org/10.7554/eLife.34408.008

• Transparent reporting form
DOI: https://doi.org/10.7554/eLife.34408.009

---

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
