## [Decision Letter]

[Editors’ note: a previous version of this study was rejected after peer review, but the authors submitted for reconsideration. The first decision letter after peer review is shown below.]

Thank you for submitting your work entitled "MR-Base: a platform for systematic causal inference across the phenome using billions of genetic associations" for consideration by *eLife*. Your article has been reviewed by two peer reviewers, one of whom is a member of our Board of Reviewing Editors and the evaluation has been overseen by a Senior Editor. The reviewers have opted to remain anonymous.

Our decision has been reached after consultation between the reviewers. Based on these discussions and the individual reviews below, we regret to inform you that your work will not be considered further for publication in *eLife*.

Both reviewers were enthusiastic about MRBase, and its value to the community. However, there were serious concerns about aspects of the manuscript (see the reviews below) that precluded further consideration of this version of the manuscript, and which went beyond the scope of what we feel is appropriate for revision. In this instance, however, should you decide to revise the manuscript along the lines recommended, we would be willing to reconsider the paper as a new submission.

Reviewer #1:

The purpose of this paper is twofold. The first part of the paper describes the MR-Base, a resource of published GWAS results and a platform that allows performing mendelian randomization (MR) analyses with the available data. In the second part of the paper, the authors perform an actually MR analysis using their MR-base platform. They assess the efficacy of lipid lowering drugs by using genetic variants that mimic the effect of those drugs (HMGCR for statins; NPC1L1 for Ezetimibe; PCSK9 for Evolocumab, APOA for Lp[a] and APOC3 for triglyceride lowering drugs) on the prevention of cardiovascular disease. In addition, the safety of the lipid-lowering drugs was assessed by testing the association with type 2 diabetes and other relevant – potentially adverse – outcomes in a hypothesis free manner. The MR analyses confirm previous (larger) MR studies and are consistent with available randomized control trials. In addition, they found evidence for potential adverse outcomes, that may have not been reported before.

The author should be commended on creating a massive resource and a useful platform for many to perform MR analyses without necessarily having immediate access to the required data. The example that was used to "pilot" the MR-Base platform is of interest, but mainly confirms what has been reported before.

My main concern is that neither the first or the second part of the paper have been sufficiently developed. For example, for the first part to be truly helpful to readers, the resource, platform and underlying methodology should have been described in more detail in the main text and more guidance should have been given at which test should/could be used under which circumstances and what their (dis)advantages are. In addition, the limitations of the MR-base compared to a typical MR analyses and compared to RCTs should have been made more explicit. Currently, this information is hidden in supplementary data.

The second part, while interesting, is mainly confirmatory and more used as an example or an application of the MR-base resource rather than a new hypothesis that is being tested. The main new observation is that MR-base allows doing a hypothesis free screen for potential (adverse/other) outcomes.

Taken together, as a reader, it may not be clear what the main aim of the paper is as neither part has great depth or innovation.

Reviewer #2:

MR-Base is an important resource and it is right that the academic community is made aware of it and that researchers have a reference that they can cite when they use it, so I am broadly supportive of this paper.

However, what the authors have produced is a non-critical description of MR-Base. There is a section entitled 'summary of limitations and some solutions' but it is only 7 lines long. The authors need to correct this imbalance.

In the supplement the authors consider 33 different examples of the use of MR-Base. If this paper is intended to inform the research community about MR-Base then one or two examples would be sufficient, provided that they include a statement to the effect that they are intended as illustrations and not as definitive research findings. It appears that the authors are looking for an easy way of laying claim to some of the obvious applications before MR-Base is made public. These applications are out of place in this paper and they are dangerous. There are so many examples they cannot be considered in the detail that one would normally find in an MR paper and just as importantly, they cannot be properly reviewed.

My suggestion is that the paper is re-written dropping most of the examples and in a style that acknowledges the limitations of the database. In that form I would support publication.

[Editors’ note: what now follows is the decision letter after the authors submitted for further consideration.]

Thank you for submitting your article "MR-Base: a database of GWAS summary data integrated with analytical tools enables causal inference across the phenome" for consideration by *eLife*. Your article has been reviewed by three peer reviewers, one of whom Ruth Loos is a member of our Board of Reviewing Editors and the evaluation has been overseen by Mark McCarthy as the Senior Editor. The following individuals involved in review of your submission have agreed to reveal their identity: Tea Skaaby (Reviewer #2); Frank Dudbridge (Reviewer #3).

The reviewers have discussed the reviews with one another and the Reviewing Editor has drafted this decision to help you prepare a revised submission. We hope you will be able to submit the revised version soon, so that we can proceed with final assessment of the paper. As usual, please provide a latter that indicates how you have responded to the comments raised.

Summary:

All three reviewers were enthusiastic and agreed that this paper (and the MR-BASE tool) are very valuable contributions to the field. We ask you to address the comments and suggestions from reviewer.

Reviewer #1:

The revised paper is very well written and provides a nice balance between the MR-base description, real-life examples and strengths and limitations. The example, estimating the causal relationship between LDL-cholesterol and coronary heart disease is informative as it also illustrates how to detect potential biases and how to resolve them. The MR-base has been an extreme valuable contribution to the field, and this paper provides the details users may need perform their own analyses.

As for the limitations; it would be great if the authors could be more explicit about how some of these limitations might affect the MR results. e.g. sample overlap could lead to "weak instrument bias", which mean what exactly?

MR-base provides meta-data to assess whether two samples differ; but how can researchers know whether differences indeed affect the MR results?

Somehow, the "sample overlap" and "two-sample assumption" are somewhat contradictory; i.e. you want the two samples to be similar, but overlap causes bias?

Reviewer #3:

Congratulations for creating a resource that should be very useful to many researchers in coming years. A huge amount of work has gone into this, saving many people a lot of time in future while reducing the chance of human error in these analyses.

I have not tested the software for this review but am aware that a user base already exists, and it is therefore ready for publication.

The article itself is well written and strikes a nice balance between describing methods, implementation and applications.

---

## [Author Response]

[Editors’ note: the author responses to the first round of peer review follow.]

Reviewer #1:The purpose of this paper is twofold. The first part of the paper describes the MR-Base, a resource of published GWAS results and a platform that allows performing mendelian randomization (MR) analyses with the available data. In the second part of the paper, the authors perform an actually MR analysis using their MR-base platform. They assess the efficacy of lipid lowering drugs by using genetic variants that mimic the effect of those drugs (HMGCR for statins; NPC1L1 for Ezetimibe; PCSK9 for Evolocumab, APOA for Lp[a] and APOC3 for triglyceride lowering drugs) on the prevention of cardiovascular disease. In addition, the safety of the lipid-lowering drugs was assessed by testing the association with type 2 diabetes and other relevant – potentially adverse – outcomes in a hypothesis free manner. The MR analyses confirm previous (larger) MR studies and are consistent with available randomized control trials. In addition, they found evidence for potential adverse outcomes, that may have not been reported before.The author should be commended on creating a massive resource and a useful platform for many to perform MR analyses without necessarily having immediate access to the required data. The example that was used to "pilot" the MR-Base platform is of interest, but mainly confirms what has been reported before.

Thank you for the kind remark, and the very valuable suggestions on making this a more focused and accessible paper.

My main concern is that neither the first or the second part of the paper have been sufficiently developed. For example, for the first part to be truly helpful to readers, the resource, platform and underlying methodology should have been described in more detail in the main text and more guidance should have been given at which test should/could be used under which circumstances and what their (dis)advantages are.

We have now completely re-written the manuscript. The first part explains the methods behind 2SMR and the assumptions they seek to address. We then describe in detail the steps that should be taken to use MR-Base. Third, we provide some examples of hypothesis-driven and hypothesis-free analyses. Finally, we discuss the strengths and limitations of the resource.

In addition, the limitations of the MR-base compared to a typical MR analyses and compared to RCTs should have been made more explicit. Currently, this information is hidden in supplementary data.

We certainly had no intention to hide the limitations section. We have now re-written the Discussion section, focussing on the limitations that MR-Base addresses, the limitations that MR-Base isn’t addressing, and new or existing weaknesses in causal inference that might be exacerbated by MR-Base.

The second part, while interesting, is mainly confirmatory and more used as an example or an application of the MR-base resource rather than a new hypothesis that is being tested. The main new observation is that MR-base allows doing a hypothesis free screen for potential (adverse/other) outcomes.Taken together, as a reader, it may not be clear what the main aim of the paper is as neither part has great depth or innovation.

We do still include an updated hypothesis-free analysis because we believe that this is absolutely necessary to illustrate the utility of the resource. With regards to innovation, we believe that the construction of a platform that integrates data with analysis on this scale is highly innovative, and we have tried to describe this in a lot of detail. We are reluctant to introduce too many new ideas in this single paper because as you have pointed out to us already, we were already struggling to be sufficiently clear. We have, however, included a brief section that describes how hypothesis-free analyses can help to understand the heterogeneity that is often observed in MR analysis – this is an entirely new way to exploit the database.

Reviewer #2:MR-Base is an important resource and it is right that the academic community is made aware of it and that researchers have a reference that they can cite when they use it, so I am broadly supportive of this paper.

We are grateful for the positive remarks about MR-Base and the valuable suggestions on improving the paper. The paper has been completely re-written to focus more on describing the resource, in terms of the problems that it solves, how to use it, showcased examples, and there is now a lengthy section on limitations.

However, what the authors have produced is a non-critical description of MR-Base. There is a section entitled 'summary of limitations and some solutions' but it is only 7 lines long. The authors need to correct this imbalance.

Thank you for this suggestion. We completely agree that it’s of paramount importance for users to be aware of the limitations of MR, and to critically evaluate any causal inference that is generated through MR or by MR-Base. As such we had previously included an extremely lengthy description of limitations in MR and MR-Base, but we deemed it too unwieldy to go anywhere other than in the Supplementary materials. We have now completely re-written the Discussion section to focus on three main topics: The limitations in MR that are addressed by MR-Base, limitations in MR that are not addressed by MR-Base, and limitations in MR that are exacerbated by MR-Base. Throughout, we draw the reader’s attention to the limitations inherent in MR and MR-Base also.

In the supplement the authors consider 33 different examples of the use of MR-Base. If this paper is intended to inform the research community about MR-Base then one or two examples would be sufficient, provided that they include a statement to the effect that they are intended as illustrations and not as definitive research findings. It appears that the authors are looking for an easy way of laying claim to some of the obvious applications before MR-Base is made public. These applications are out of place in this paper and they are dangerous. There are so many examples they cannot be considered in the detail that one would normally find in an MR paper and just as importantly, they cannot be properly reviewed.My suggestion is that the paper is re-written dropping most of the examples and in a style that acknowledges the limitations of the database. In that form I would support publication.

Thank you for the suggestion. Please be assured that we had no intention to lay claim to a large number of results through this paper – we were simply trying to demonstrate the utility of the resource. The paper has now been re-written to focus much more on describing the resource, and throughout we emphasise that the results presented are there to showcase the resource, and in general any results obtained through MR analysis should be triangulated with other experimental designs where possible. We do believe it is important to show-case the utility of MR-Base for both hypothesis-driven and hypothesis-free causal inference, and so have retained these analyses. We have omitted the follow up graphs for those suggestive findings in order to maintain focus on describing the resource, and we now flag in the paper that any putative associations need to be followed up with dedicated analysis in separate studies.

[Editors' note: the author responses to the re-review follow.]

Reviewer #1:The revised paper is very well written and provides a nice balance between the MR-base description, real-life examples and strengths and limitations. The example, estimating the causal relationship between LDL-cholesterol and coronary heart disease is informative as it also illustrates how to detect potential biases and how to resolve them. The MR-base has been an extreme valuable contribution to the field, and this paper provides the details users may need perform their own analyses.As for the limitations; it would be great if the authors could be more explicit about how some of these limitations might affect the MR results. e.g. sample overlap could lead to "weak instrument bias", which mean what exactly?

We have provided further clarifications and examples of impacts to text (see Discussion section). In the example given by the reviewer, sample overlap could bias associations towards the confounded observational association – a phenomenon known as weak instrument bias. Bias from sample overlap can, however, be minimized by using strong instruments (e.g. an F statistic much greater than 10 for the instrument-exposure association). If the overlapping samples also include the discovery study this can compound the problem. This can be avoided by using replication samples to define instrument-exposure effects.

MR-base provides meta-data to assess whether two samples differ; but how can researchers know whether differences indeed affect the MR results?

Clarifications added to Discussion section. If the populations are different for the exposure and outcome studies (e.g. European vs East Asian study), it is likely that this will lead to at least some mis-estimation of the magnitude of the association between exposure and outcome, although inferences about directions of causality should remain unbiased. If using samples from different populations is unavoidable, users should acknowledge the impact of this limitation and restrict their conclusions to directions of causality.

Somehow, the "sample overlap" and "two-sample assumption" are somewhat contradictory; i.e. you want the two samples to be similar, but overlap causes bias?

Clarifications added to Discussion section. The exposure and outcome studies should not involve overlapping participants (i.e. the participating individuals should not be members of both studies) but the participants from both studies should come from the same popuaslation – i.e. they should be of similar age and sex distribution and come from the same geographic area. In addition, they should have similar patterns of LD in the genomic regions used to define the instruments (i.e. come from similar genetic ancestry).